# Principal Components Bias in Deep Neural Networks

## Abstract

Recent work suggests that convolutional neural networks of different architectures learn to classify images in the same order. To understand this phenomenon, we revisit the over-parametrized deep linear network model. Our asymptotic analysis, assuming that the hidden layers are wide enough, reveals that the convergence rate of this model's parameters is exponentially faster along directions corresponding to the larger principal components of the data, at a rate governed by the singular values. We term this convergence pattern the *Principal Components bias (PC-bias)*. We show how the *PC-bias* streamlines the order of learning of both linear and non-linear networks, more prominently at earlier stages of learning. We then compare our results to the spectral bias, showing that both biases can be seen independently, and affect the order of learning in different ways. Finally, we discuss how the *PC-bias* may explain some benefits of early stopping and its connection to PCA, and why deep networks converge more slowly when given random labels.

## 1 Introduction

The dynamics of learning in deep neural networks is an intriguing subject, not yet sufficiently understood. Diverse empirical data seems to support the hypothesis that neural networks start by learning a simple model, which then gains complexity as learning proceeds (Gunasekar et al., 2018; Soudry et al., 2018; Hu et al., 2020; Nakkiran et al., 2019; Gissin et al., 2019; Heckel & Soltanolkotabi, 2019; Ulyanov et al., 2018; Valle-Perez et al., 2018). This phenomenon is sometimes called *simplicity bias* (Dingle et al., 2018; Shah et al., 2020).

Recent work additionally shows that neural networks learn the training examples of natural datasets in a consistent order, and further impose a consistent order on the test set (Hacohen et al., 2020; Pliushch et al., 2021). Below we call this effect *Learning Order Constancy* (LOC). Currently, the characteristics of visual data, which may explain this consistently imposed order, remain unclear. Surprisingly, this universal order persists despite the variability introduced into the training of different models and architectures.

To understand this phenomenon, we start by analyzing the deep linear network model (Saxe et al., 2013, 2019), defined by the concatenation of linear operators. While not a universal approximator, it is nevertheless trained by minimizing a non-convex objective function with a multitude of minima. The investigation of such networks is often employed to shed light on the learning dynamics when complex geometric landscapes are explored by GD (Fukumizu, 1998; Arora et al., 2018).

In Section 2, we prove that the convergence of the weights of deep linear networks is governed by the eigendecomposition of the raw data in a phenomenon we term *PC-bias*. These asymptotic results, valid when the hidden layers are wide enough, can be seen as an extension of the known behavior of the single-layer convex linear model (Le Cun et al., 1991). Our work is closely related to (Saxe et al., 2013, 2019), where the deep linear model's dynamics is analyzed as a function of the input and input-output statistics. Importantly, the analysis in (Saxe et al., 2013, 2019; Arora et al.,

2018) incorporates the simplifying assumption that the data's singular values are identical (whitened data), an assumption which unfortunately **obscures the main result of our analysis** – the direct dependence of convergence rate on the singular values of the data.

In Section 3, we empirically show that this pattern of convergence is indeed observed in deep linear networks, validating the plausibility of our assumptions. We continue by showing that the *LOC-effect* in deep linear network is determined solely by their *PC-bias*. We prove a similar (weaker) result for the non-linear two-layer ReLU model introduced by Allen-Zhu et al. (2018), where this model is presented as a certain extension of NTK (Jacot et al., 2020). In this framework, convergence is fastest along the largest **kernel's** principal components, a result related to the *Spectral bias* discussed below.

In Section 4, we extend the study empirically to non-linear networks, and investigate the relation between the *PC-bias* and the *LOC-effect* in general deep networks. We first show that the order by which examples are learned by linear networks is highly correlated with the order induced by prevalent deep CNN models. We then show directly that the learning order of non-linear CNN models is affected by the principal decomposition of the data. Moreover, the *LOC-effect* diminishes when data is whitened, indicating a tight connection between the *PC-bias* and the *LOC-effect*.

Our results are reminiscent of another phenomenon, termed *Spectral bias* (Rahaman et al., 2019; Cao et al., 2019), which associates the learning dynamics of neural networks with the Fourier decomposition of functions in the hypothesis space. Rahaman et al. (2019) empirically demonstrated that the complexity of classifiers learned by ReLU networks increases with time. Basri et al. (2019, 2020) showed theoretically, by way of analyzing elementary neural network models, that these models first fit the data with low-frequency functions, and gradually add higher frequencies to improve the fit.

Nevertheless, the *spectral bias* and *PC-bias* are inherently different. Indeed, the eigendecomposition of raw images is closely related to the Fourier analysis of images as long as the statistical properties of images are (approximately) translation-invariant (Simoncelli & Olshausen, 2001; Torralba & Oliva, 2003). Still, the *PC-bias* is guided by spectral properties of the raw data and is additionally blind to class labels. On the other hand, the *spectral bias*, as well as the related *frequency bias* that has been shown to characterize NTK models (Basri et al., 2020), are all guided by spectral properties of the learned hypothesis, which strongly depends on label assignment.

In Section 4.3 we investigate the relation between the *PC-bias*, *spectral bias*, and the *LOC-effect*. We find that the *LOC-effect* is very robust: (i) when we neutralize the *spectral bias* by using low complexity models such as deep linear networks, the effect is still observed; (ii) when we neutralize the *PC-bias* by using whitened data, the *LOC-effect* persists. We hypothesize that at the beginning of learning, the learning dynamics of neural models is controlled by the eigendecomposition of the raw data. As learning proceeds, control of the dynamics slowly shifts to other factors.

The PC-bias has implications beyond the *LOC-effect*, as expanded in Section 5 and Suppl. §A:

**1. Early stopping.** It is often observed that when training deep networks with real data, the highest generalization accuracy is obtained before convergence. Consequently, early stopping is often prescribed to improve generalization. Following the commonly used assumption that in natural images the lowest principal components correspond to noise (Torralba & Oliva, 2003), our results predict the benefits of early stopping, and relate it to PCA. In Section 5 we investigate the relevance of this conclusion to real non-linear networks (see, e.g., Basri et al. (2019); Li et al. (2020) for complementary accounts).

**2. Slower convergence with random labels.** Zhang et al. (2016) showed that neural networks can learn any label assignment. However, training with random label assignments is known to converge slower as compared to training with the original labels (Krueger et al., 2017). We report a similar phenomenon when training deep linear networks. Our analysis shows that when the principal eigenvectors are correlated with class identity, as is often the case in natural images, the loss decreases faster when given true label assignments as against random label assignments. In Section 5 we investigate this hypothesis empirically in linear and non-linear networks.

**3. Weight initialization.** Different weight initialization schemes have been proposed to stabilize the learning and minimize the hazard of "exploding gradients" (e.g., Glorot & Bengio, 2010; He et al., 2015). Our analysis (see Suppl. §A) identifies a related variant, which eliminates the hazard when all the hidden layers are roughly of equal width. In the deep linear model, it can be proven that the proposed normalization variant in a sense minimizes repeated gradient amplification.

## 2  Theoretical analysis

**Notations.** Let $\mathbb{X} = \{(\boldsymbol{x}_i, \boldsymbol{y}_i)\}_{i=1}^n$ denote the training data, where $\boldsymbol{x} \in \mathbb{R}^q$ denotes the i-th data point and $\boldsymbol{y} \in \{0, 1\}^K$ its corresponding label. Let $\frac{1}{n_i}\boldsymbol{m}_i$ denote the centroid (mean) of class $i$ with $n_i$ points, and $M = [\boldsymbol{m}_1 \ldots \boldsymbol{m}_K]^\top$. Finally, let $X$ and $Y$ denote the matrices whose $i^{th}$ column is $\boldsymbol{x}_i$ and $\boldsymbol{y}_i$ respectively. $\Sigma_{XX} = XX^\top$ and $\Sigma_{YX} = YX^\top$ denote the covariance matrix of $X$ and cross-covariance of $X$ and $Y$ respectively. We note that $\Sigma_{XX}$ captures the structure of the data irrespective of class identity.

**Definition 1** (Principal coordinate system)**.** *The coordinate system obtained by rotating the data in $\mathbb{R}^q$ by an orthonormal matrix $U^\top$, where $SVD(\Sigma_{XX}) = UDU^\top$. Now $\Sigma_{XX} = D$, a diagnoal matrix whose elements are the singular values of $XX^\top$, arranged in decreasing order $d_1 \geq d_2 \geq \ldots \geq d_q \geq 0$.*

**Definition 2** (Compact representation)**.** *Let $f(\boldsymbol{x})$ denote a deep linear network. Then $f(\boldsymbol{x}) = \left(\prod_{l=L}^1 W_l\right)\boldsymbol{x} = \boldsymbol{W}\boldsymbol{x}$, where $\boldsymbol{W} \in \mathbb{R}^{K \times q}$ is called the compact representation of the network.*

**Definition 3** (Error matrix)**.** *For a deep linear network whose compact representation is $\boldsymbol{W}$, the error matrix is $Er = \boldsymbol{W}\Sigma_{XX} - \Sigma_{YX}$. In the principal coordinate system, $Er = \boldsymbol{W}D - M$.*

**Assumptions.** Our analysis assumes that the learning rate $\mu$ is infinitesimal, and therefore terms of size $O(\mu^2)$ can be neglected. We further assume that the width of the hidden layers lies in $[\mathrm{m}, \mathrm{m} + M_b]$, where $\mathrm{m} \to \infty$ denotes a very large number and $M_b$ is fixed. Thus terms of size $O(\frac{1}{m})$ can also be neglected. In Fig. 1 we show the plausibility of these assumptions, where the predicted dynamics is seen throughout the training of deep linear networks, even for small values of $\mathrm{m}$.

### 2.1  The dynamics of deep over-parametrized linear networks

Consider a deep linear network with $L$ layers, and let

$$L(\mathbb{X}) = \frac{1}{2}\|\boldsymbol{W}X - Y\|_F^2 \qquad \boldsymbol{W} := \prod_{l=L}^1 W_l, \quad W_l \in \mathbb{R}^{m_l \times m_{l-1}} \tag{1}$$

Above $m_l$ denotes the number of neurons in layer $l$, where $m_0 = q$ and $m_L = K$.

**Theorem 1.** *In each time point $s$, the compact matrix representation $\boldsymbol{W}$ obeys the following dynamics, when using the notation $Er^s$ defined in Def. 3:*

$$\boldsymbol{W}^{s+1} = \boldsymbol{W}^s - \mu \sum_{l=1}^L A_l^s \cdot Er^s \cdot B_l^s + O(\mu^2) \tag{2}$$

*Above $\mu$ denotes the learning rate. $A_l^s$ and $B_l^s$ are called* gradient scale matrices*, and are defined as*

$$A_l^s := \left(\prod_{j=L}^{l+1} W_j^s\right)\left(\prod_{j=L}^{l+1} W_j^s\right)^\top \in \mathbb{R}^{K \times K} \qquad B_l^s := \left(\prod_{j=l-1}^1 W_j^s\right)^\top\left(\prod_{j=l-1}^1 W_j^s\right) \in \mathbb{R}^{q \times q} \tag{3}$$

The proof can be found in Suppl. §B.

**Gradient scale matrices.** Some statistical properties of such matrices are established in Suppl. §A. Note that when the number of hidden layers is $0$ ($L = 1$), both gradient scale matrices reduce to the identity matrix and the dynamics in (2) is reduced to the following known result (e.g., Le Cun et al., 1991): $\boldsymbol{W}^{s+1} = \boldsymbol{W}^s - \mu Er^s$. Recall, however, that the focus of this paper is the over-parameterized linear model with $L > 1$, in which the loss is not convex. Since the difference between the convex linear model and the over-parameterized deep model boils down to these matrices, our convergence analysis henceforth focuses on the dynamics of the *gradient scale matrices*.

In accordance, we analyze the evolution of the *gradient scale matrices* as learning proceeds. Let $\mathrm{m} = \min(m_1, ..., m_{L-1})$ denote the size of the smallest hidden layer. Initially for $s = 0$, all weight matrices $W_l^0$ are assumed to be initialized by sampling from a distribution with mean $0$ and variance $\sigma_l^2 = O(\frac{1}{\mathrm{m}})$. The specific normalization factor, alluded to in $O(\frac{1}{\mathrm{m}})$, is a variant of the Glorot initialization. Details and justification can be found in Suppl. §A.1.

At time $s$, let $A_l^s(\mathrm{m})$ and $B_l^s(\mathrm{m})$ denote a sequence of random *gradient scale matrices*, corresponding to networks whose smallest hidden layer has $\mathrm{m}$ neurons. From Suppl. §A we deduce that:

**Theorem 2.** *Using $\xrightarrow{p}$ to denote convergence in probability as $\mathtt{m} \to \infty$, and $\forall s, l$:*

$$B_l^s(\mathtt{m}) \xrightarrow{p} I, \quad \mathrm{var}[B^l(\mathtt{m})] = O\left(\frac{1}{\mathtt{m}}\right) \qquad A_l^s(\mathtt{m}) \xrightarrow{p} I, \quad \mathrm{var}[A^l(\mathtt{m})] = O\left(\frac{1}{\mathtt{m}}\right)$$

*Proof.* Proof by induction on $s$. Initially when $s = 0$, the claim follows from Thm 4 and Corr 5.1. The induction step validity follows from Thm 6 and Thm 7 (see Suppl. §A.2). $\qquad\square$

The detailed proof shows that the relevant constants are amplified with $s$. While they remain moderate and $\mathtt{m}$ is sufficiently large, $B_l^s(\mathtt{m}) \approx I$ and $A_l^s(\mathtt{m}) \approx I \; \forall l$. In this case, the dynamics of the over-parameterized model is identical to the dynamics of the convex linear model, $\boldsymbol{W}^{s+1} = \boldsymbol{W}^s - \mu E r^s$.

**Convergence rate.** In §A.2 we show that the convergence of $B_l^s(\mathtt{m})$ to $I$ is governed to some extent by $O\left(\frac{K}{\mathtt{m}}\right)$, while the convergence of $A_l^s(\mathtt{m})$ is governed by $O\left(\frac{q}{\mathtt{m}}\right)$. Recall that while $\mathtt{m} \to \infty$, $q$ is the dimension of the data space which is fixed in advance and can be fairly large, while $K$ is the number of classes which is fixed and quite small. Typically, $K \ll q$. Thus we expect the right *gradient scale matrices* $B_l^s(\mathtt{m})$ to remain approximately $I$ much longer than the left matrices $A_l^s(\mathtt{m})$.

**Empirical validation.** Since the results above are asymptotic, and to envision the difference between convergence governed by $O\left(\frac{K}{\mathtt{m}}\right)$ vs. $O\left(\frac{q}{\mathtt{m}}\right)$, we resort to simulations whose results are shown in Fig. 1. These empirical results, recounting linear networks with 4 hidden layers of width 1024, clearly show that during a significant part of the training both *gradient scale matrices* remain approximately $I$. The difference between the convergence rate of $B_l^s$ and $A_l^s$ is seen later on, when $\Delta A_l^s$ starts to increase shortly before convergence, while $\Delta B_l^s$ remains essentially 0 throughout.

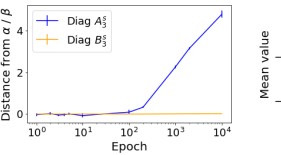 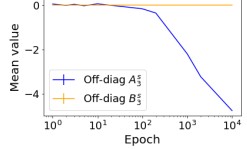

(a) Diagonal layer 3     (b) Off-diagonal layer 3

Figure 1: The dynamics of $A_l^s$ and $B_l^s$ when training 10 5-layered linear networks on the small-mammals dataset. (a) Mean distance of the diagonal elements of $A_l^s$ and $B_l^s$ from $\alpha_i^s$ and $\beta_i^s$ (as defined in Thm 3, §A.1). (b) Mean value of the off-diagonal elements of $A_l^s$ and $B_l^s$. The networks reach maximal test accuracy at epoch $s = 100$, before the divergence of $A_l^s$. All layers behave similarly.

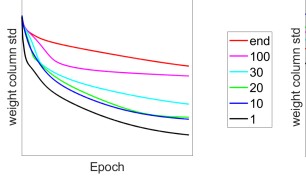 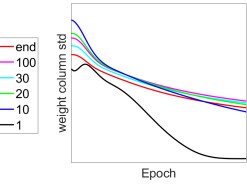

(a) Linear Network     (b) ReLU Network

Figure 2: Empirical confirmation of the theoretical results reported below, showing the std of $\boldsymbol{w}_j$ over 10 independently trained networks as a function of the epoch, for 6 specific principal components (identified in the legend). Left: two-layer linear network. Right: two-layer non-linear network with ReLU activation.

## 2.2 Weight evolution

$K \ll q$ entails that $B_l^s(\mathtt{m})$ remains approximately equal to $I$ much longer than $A_l^s(\mathtt{m})$. This is substantiated by the simulation results in Fig. 1. Consequently, while earlier on it is safe to assume that both $A_l^s \approx I$ and $B_l^s \approx I$, as learning proceeds only $B_l^s \approx I$ is safe to assume.

With this in mind, we obtain expressions for the evolution of $\boldsymbol{W}^s$ separately for earlier and later in learning. We first shift to the principal coordinate system defined in Def 1. In this system we can analyze each column of $\boldsymbol{W}^s$ separately, where $\boldsymbol{w}_j^s$ and $\boldsymbol{m}_j$ denote the respective columns of $\boldsymbol{W}^s$ and $M$. At the beginning of learning when both $A_l^s \approx I$ and $B_l^s \approx I$ (see §B.3 for a detailed derivation):

$$\boldsymbol{w}_j^{s+1} = (\lambda_j)^s \boldsymbol{w}_j^0 + [1 - (\lambda_j)^s]\frac{\boldsymbol{m}_j}{d_j} \qquad\qquad \lambda_j = 1 - \mu d_j L \qquad\qquad (4)$$

Eq. 4 is reminiscent of the well understood dynamics of training the convex one layer linear model. It is composed of two additive terms, revealing two parallel and independent processes:

     1. The dependence on random initialization tends to 0 exponentially with decline rate $\lambda_j$.

     2. The final value is the sum of a geometrical series with a common ratio $\lambda_j$.

In either case, convergence is fastest for the largest singular eigenvalue, or the first column of $\boldsymbol{W}$, and slowest for the smallest singular value. This behavior is visualized in Fig. 2a. Importantly, the rate of convergence depends on the singular value $d_j$, the number of layers $L$, and the learning rate $\mu$.

In later stages of learning, when we can only assume that $B_l^s \approx I$, the dynamic becomes:

$$\boldsymbol{w}_j^{s+1} = \prod_{\nu=1}^{s}(I - \mu d_j A^\nu)\boldsymbol{w}_j^0 + \mu \left[\sum_{\nu=1}^{s}\prod_{\rho=\nu+1}^{s}(I - \mu d_j A^\rho)A^\nu\right]\boldsymbol{m}_j \tag{5}$$

where $A^s = \sum_{l=1}^{L} A_l^s$. The proof is provided in §B.3. Although the dynamics now depends on matrices $A^s$ as well, it is still the case that the convergence of each column is governed by its singular value $d_j$. This suggests that while the *PC-bias* is more pronounced in earlier stages of learning, its effect persists throughout.

The analysis above is extended to a simple non-linear ReLU model (cf. Arora et al., 2019) as detailed in §B.2, with qualitatively similar results (albeit under unrealistic assumptions). Empirical results, shown in Fig. 2b, indicate that the results are indicative beyond the assumed circumstances.

## 3 PC-bias: empirical study

In this section, we first analyze deep linear networks, showing that the convergence rate is indeed governed by the principal singular values of the data, which demonstrates the plausibility of the assumptions made in Section 2. We continue by extending the scope of the investigation to non-linear neural networks, finding there evidence for the *PC-bias* mostly in the earlier stages of learning.

### 3.1 Methodology

We say that a linear network is $L$-layered when it has $L - 1$ hidden fully connected (FC) layers (without convolutional layers). In our empirical study we relaxed some assumptions of the theoretical study, in order to increase the resemblance of the trained networks to networks in common use. Specifically, we changed the initialization to the commonly used Glorot initialization, replaced the $L_2$ loss with the cross-entropy loss, and employed SGD instead of the deterministic GD. Notably, the original assumptions yielded similar results. The results presented summarize experiments with networks of equal width across all hidden layers, specifically the moderate value of $\mathtt{m} = 1024$, keeping in mind that we test the relevance of asymptotic results for $\mathtt{m} \to \infty$. Using a different width for each layer yielded similar qualitative results. Details regarding the hyper-parameters, architectures, and datasets can be found in §D.1, §D.3 and §D.4 respectively.

### 3.2 PC-bias in deep linear networks

In this section, we train $L$-layered linear networks, then compute their compact representations $\boldsymbol{W}$ rotated to align with the canonical coordinate system (Def. 1). Note that each row $\boldsymbol{w}_r$ in $\boldsymbol{W}$ essentially defines the one-vs-all separating hyper-plane corresponding to class $r$.

To examine both the variability between models and their convergence rate, we inspect $\boldsymbol{w}_r$ at different time points during learning. The rate of convergence can be measured directly, by observing the changes in the weights of each element in $\boldsymbol{w}_r$. These weight values[1] should be compared with the optimal values in each row $\boldsymbol{w}_r$ of $W_{opt} = YX^T(XX^T)$. The variability between models is measured by calculating the standard deviation (std) of each $\boldsymbol{w}_r$ across $N$ models.

We begin with linear networks. We trained 10 5-layered FC linear networks, and 10 linear st-VGG convolutional networks. When analyzing the compact representation of such networks we observe similar behavior – weights corresponding to larger principal components converge faster to the optimal value, and their variability across models converges faster to 0 (Figs. 3a,3b). Thus, while the theoretical results are asymptotic, *PC-bias* is empirically seen throughout the entire learning process of deep linear networks.

**Whitened data.** The *PC-bias* is neutralized when the data is whitened, at which point $\Sigma_{XX}$ is the scaled identity matrix. In Fig. 3c, we plot the results of the same experimental protocol while using a ZCA-whitened dataset. As predicted, the networks no longer show any bias towards any principal direction. Weights in all directions are scaled similarly, and the std over all models is the same in each epoch, irrespective of the principal direction. (Additional experiments show that this is *not* an artifact of the lack of uniqueness when deriving the principal components of a white signal).

---

[1]We note that the weights tend to start larger for smaller principal components, as can be seen in Fig. 3a left.

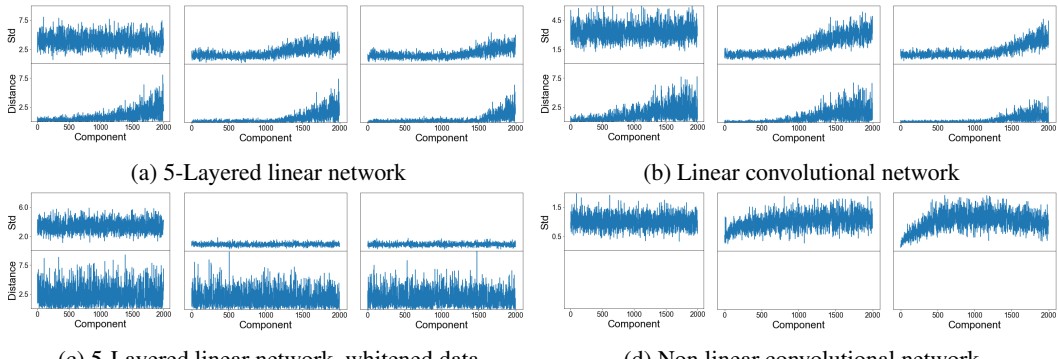

(a) 5-Layered linear network  (b) Linear convolutional network

(c) 5-Layered linear network, whitened data  (d) Non linear convolutional network

Figure 3: Convergence of the compact representation along the principal directions in different epochs. The value of the $X$-axis corresponds to the index of a principal eigenvalue, from the most significant to the least significant. (a) 10 5-layered linear networks trained on the cats and dogs dataset. 3 plots are provided, corresponding to snapshots taken at different stages of learning: the beginning (epoch 0, left), intermediate stage (middle), and close to convergence (right). Bottom panel: average distance of the weights in $\boldsymbol{w}_1$ from the optimal linear classifier; top panel: respective std. (b) Similarly, for 10 linear st-VGG convolutional networks, trained on CIFAR-10. (c) Similarly, for 10 5-layered linear networks, trained on the cats and dogs dataset, with ZCA-whitening. (d) Similarly, for 10 **non-linear** st-VGG networks trained on the cats and dogs dataset. Here the distance to the optimal solution is not well defined and we therefore only show the std.

### 3.3  PC-bias in general CNNs

In this section, we investigate the manifestation of the *PC-bias* in non-linear deep convolutional networks. As we cannot directly track the learning dynamics separately in each principal direction of non-linear networks, we adopt two different evaluation mechanisms:

**Linear approximation.** We considered several linear approximations, but since all of them showed the same qualitative behavior, we report results with the simplest one. Specifically, to obtain a linear approximation of a non-linear network, without max-pooling or batch-normalization layers, we follow the definition of the compact representation from Section 2 while ignoring any non-linear activation. We then align this matrix with the canonical coordinate system (Def. 1), and observe the evolution of the weights and their std across models along the principal directions during learning. Note that now the networks do not converge to the same compact representation, which is not unique. Nevertheless, we see that the *PC-bias* governs the weight dynamics to a noticeable extent.

More specifically, in these networks a large fraction of the lowest principal components hardly changes during learning, as good as being ignored. Nevertheless, the *PC-bias* affects the higher principal components, most notably at the beginning of training (see Fig. 3d). Thus weights corresponding to higher principal components converge faster, and the std across models of such weights decreases faster for higher principal components.

**Projection to higher PC's.** We created a modified *test-set*, by project-ing each test example on the span of the first $P$ principal components. This is equivalent to reducing the dimensionality of the test set to $P$ us-ing PCA. We trained an ensemble of $N$=100 st-VGG networks on the original small mammals training set, then evaluated these networks dur-ing training on 4 versions of the test-set, reduced to $P$=1,10,100,1000 dimensions respectively. Mean accuracy is plotted in Fig. 4. Similar results are obtained when training VGG-19 networks on CIFAR-10, see §C.3.

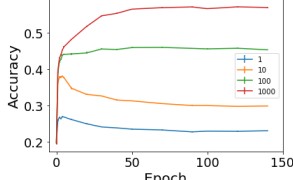

Figure 4: Mean accuracy of 10 st-VGG networks evaluated on test data projected to dimen-sionality $\{1, 10, 100, 1000\}$.

Taking a closer look at Fig. 4, we see that when evaluated on lower dimensionality test-data ($P$=1,10), the networks' accuracy peaks after a few epochs, at which point performance starts to decrease. This result suggests that the networks rely more heavily on these dimensions in the earlier phases of learning, and then continue to learn other things. In contrast, when evaluated on higher dimensionality test-data ($P$=100,1000), accuracy continues to rise, longer so for larger $P$. This suggests that significant learning of the additional dimensions continues in later stages of the learning.

## 4 PC-bias: Learning Order Constancy

In this section, we show that the *PC-bias* is significantly correlated with the learning order of deep neural networks, and can therefore partially account for the *LOC-effect* described in Section 1. Following Hacohen et al. (2020), we measure the "speed of learning" of each example by computing its *accessibility* score. This score is given per example, and characterizes how fast an ensemble of $N$ networks learns it. Formally, $accessibility(\boldsymbol{x}) = \mathbb{E}\left[\mathbb{1}(f_i^e(\boldsymbol{x}) = y(\boldsymbol{x}))\right]$, where $f_i^e(\boldsymbol{x})$ denotes the outcome of the $i$-th network trained over $e$ epochs, and the mean is taken over networks and epochs. For the set of datapoints $\{(\boldsymbol{x}_j, \boldsymbol{y}_j)\}_{j=1}^n$, *Learning Order Constancy* is manifested by the high correlation between 2 instances of $accessibility(\boldsymbol{x})$, each computed from a different ensemble.

*PC-bias* is shown to pertain to *LOC* in two ways: First, in Section 4.1 we show high correlation between the learning order in deep linear and non-linear networks. Since the *PC-bias* fully accounts for *LOC* in deep linear networks, this suggests it also accounts (at least partially) for the observed *LOC* in non-linear networks. Comparison with the *critical principal component* verifies this assertion. Second, we show in Section 4.2 that when the *PC-bias* is neutralized, *LOC* diminishes as well. In Section 4.3 we discuss the relationship between the *spectral bias*, *PC-bias* and the *LOC-effect*.

### 4.1 PC-Bias is correlated with LOC

We first compare the order of learning of non-linear models and deep linear networks by computing the correlation between the *accessibility* scores of both models. This comparison reveals high correlation ($r = 0.85$, $p < 10^{-45}$), as seen in Fig. 5a. To investigate directly the connection between the *PC-bias* and *LOC*, we define the *critical principal component* of an example to be the first principal component $P$, such that a linear classifier trained on the original data can classify the example correctly when projected to $P$ principal components. We trained $N$=100 st-VGG networks on the cats and dogs dataset, and computed for each example its *accessibility* score and *critical principal component*. In Fig. 5b we see strong negative correlation between the two scores ($p$=$-0.93$, $r < 10^{-4}$), suggesting that the *PC-bias* affects the order of learning as measured by *accessibility*.

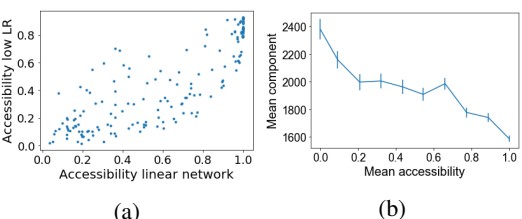

(a)             (b)

Figure 5: (a) Correlation between the *accessibility* score of $N$=100 st-VGG networks trained with a low learning rate[2], and $N$=100 linear st-VGG networks, trained on small mammals. (b) Correlation between the accessibility score of $N$=100 st-VGG networks trained on cats and dogs, and the *critical principal component* score. The *accessibility* plot is smoothed by moving average of width 10. Error bars indicate standard error.

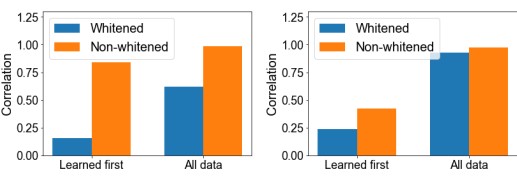

(a) Linear networks      (b) Non-linear networks

Figure 6: *LOC* measured with and without *PC-bias*. Each bar represents the correlation between the learning order of 2 collections of 10 networks trained on CIFAR-10. Orange bars represent natural images, in which the *PC-bias* is present, while blue bars represent whitened data, in which the *PC-bias* is eliminated. As *PC-bias* is more prominent earlier on, we compare these correlations for the entire data (right 2 bars), and for the subset of 20% "fastest learned" examples (left 2 bars).

### 4.2 Neutralizing the PC-bias leads to diminishing LOC

Whitening the data eliminates the *PC-bias* as shown in Fig. 3c, since all the singular values are now identical. Here we use this observation to further probe into the dependency of the *Learning Order Constancy* on the *PC-bias*. Starting with the linear case, we train 4 ensembles of $N$=10 2-layered linear networks on the cats and dogs dataset, 2 with and 2 without ZCA-whitening. We compute the *accessibility* score for each ensemble separately, and correlate the scores of the 2 ensembles in each test case. Each correlation captures the consistency of the *LOC-effect* for the respective condition. This correlation is expected to be very high for natural images. Low correlation implies that the *LOC-effect* is weak, as training the same network multiple times yields a different learning order.

---

[2]As non-linear models achieve the accuracy of linear models within an epoch or 2, low learning rate is used.

Fig. 6a shows the results for deep linear networks. As expected, the correlation when using natural images is very high. However, when using whitened images, correlation plummets, indicating that the *LOC-effect* is highly dependent on the *PC-bias*. We note that the drop in the correlation is much higher when considering only the 20% "fastest learned" examples, suggesting that the *PC-bias* affects learning order more evidently at earlier stages of learning.

Fig. 6b shows the results when repeating this experiment with non-linear networks, training 2 collections of $N$=10 VGG-19 networks on CIFAR-10. We find that the elimination of the *PC-bias* in this case affects *LOC* much less, suggesting that the *PC-bias* can only partially account for the *LOC-effect* in the non-linear case. However, we note that at the beginning of learning, when the *PC-bias* is most pronounced, once again the drop is much larger and very significant (half).

### 4.3 Spectral bias, PC-bias and LOC

The *spectral bias* (Rahaman et al., 2019) characterizes the dynamics of learning in neural networks differently, asserting that initially neural models can be described by low frequencies only. This may provide an alternative explanation to LOC. Recall that LOC is manifested in the consistency of the *accessibility* score across networks. To compare between the *spectral bias* and *accessibility* score, we first need to estimate for each example whether it can be correctly classified by a low frequency model. Accordingly, we define for each example a *discriminability* measure – the percentage out of its $k$ neighbors that share with it class identity. Intuitively, an example has a low *discriminability* score when it is surrounded by examples from other classes, which forces the learned boundary to incorporate high frequencies. In §C.2 we show that in the 2D case analyzed by Rahaman et al. (2019), this measure strongly correlates ($r$=$-0.8$, $p < 10^{-2}$) with the spectral bias.

We trained several networks (VGG-19 and st-VGG) on several real datasets, including small-mammals, STL-10, CIFAR-10/100 and a subset of ImageNet-20. For each network and dataset, we computed the *accessibility* score as well as the *discriminability* of each example. The vector space, in which discriminability is evaluated, is either the raw data or the network's perceptual space (penultimate layer activation). The correlation between these scores is shown in Table 1.

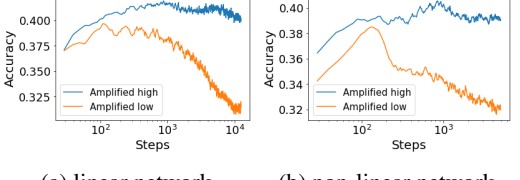

(a) linear network    (b) non-linear network

Figure 7: Effects of amplifying the highest (blue) and lowest (orange) principal components.

Table 1: Correlation between *accessibility* and *discriminability*.

| Dataset | Raw data | Penultimate |
|---|---|---|
| Small mammals | 0.46 | 0.85 |
| ImageNet 20 | 0.01 | 0.54 |
| CIFAR-100 | 0.51 | 0.85 |
| STL10 | 0.44 | 0.7 |

Using raw data, low correlation is still seen between the *accessibility* and *discriminability* scores when inspecting the smaller datasets (small mammals, CIFAR-100 and STL10). This correlation vanishes when considering the larger ImageNet-20 dataset. It would appear that on its own, the *spectral bias* cannot adequately explain the *LOC-effect*. On the other hand, in the perceptual space, the correlation between *discriminability* and *accessibility* is quite significant for all datasets. Contrary to our supposition, it seems that networks learn a representation where the *spectral bias* is evident, but this bias does not necessarily govern its learning before the representation has been learned.

## 5 PC-bias: further implications

**Early Stopping and the Generalization Gap.** Considering natural images, it is often assumed that the least significant principal components of the data represent noise (Torralba & Oliva, 2003). In such cases, our analysis predicts that as noise dominates the components learned later in learning, early stopping is likely to be beneficial. To test this hypothesis directly, we manipulated CIFAR-10 to amplify the signal in either the 1.5% most significant (higher) or 1.5% least significant (lower) principal components (see examples in Fig. 16, Suppl. §D). Accuracy over the original test set, after training 10 st-VGG and linear st-VGG networks on these manipulated images, can be seen in Fig. 7. Both in linear and non-linear networks, early stopping is more beneficial when lower

principal components are amplified, and significantly less so when higher components are amplified, as predicted by the *PC-bias*.

**Slower Convergence with Random Labels.** Deep neural models can learn any random label assignment to a given training set (Zhang et al., 2016). However, when trained on randomly labeled data, convergence appears to be much slower (Krueger et al., 2017). Assume, as before, that in natural images the lower principal components are dominated by noise. We argue that the *PC-bias* now predicts this empirical result, since learning randomly labeled examples requires signal present in lower principal components. To test this hypothesis directly, we trained 10 2-layered linear networks on datasets of natural images. Indeed, these networks converge slower with random labels (see Fig. 8a). In Fig. 8b we repeat this experiment after having whitened the images, to neutralize the *PC-bias*. Now convergence rate is identical, whether the labels are original or shuffled. Clearly, in deep linear networks the *PC-bias* gives a full account of this phenomenon.

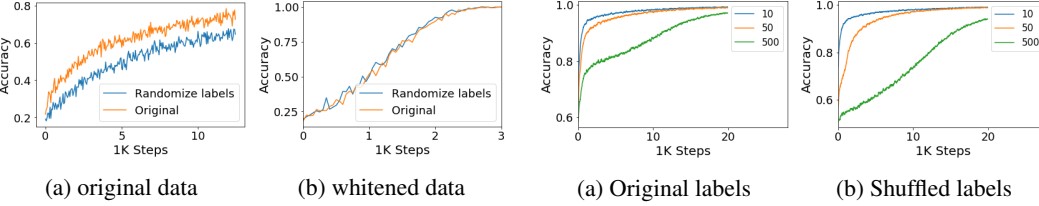

| (a) original data | (b) whitened data | (a) Original labels | (b) Shuffled labels |

Figure 8: Learning curves of 10 2-layered linear networks, with real and shuffled labels, (a) before and (b) after whitening.

Figure 9: Learning curves of st-VGG networks trained on 3 datasets, which are linearly separable after projection to the highest $P$ principal components (see legend).

To further check the relevance of this account to non-linear networks, we artificially generate datasets where only the first $P$ principal components are discriminative, while the remaining components become noise by design. We constructed two such datasets: in one the labels are correlated with the original labels, in the other they are not. Specifically, PCA is used to reduce the dimensionality of a two-class dataset to $P$, and the optimal linear separator in the reduced representation is computed. Next, all the labels of points that are incorrectly classified by the optimal linear separator are switched, so that the train and test sets are linearly separable by this separator. Note that the modified labels are still highly correlated with the original labels (for $P = 500$: $p = 0.82$, $r < 10^{-10}$). The second dataset is generated by repeating the process while starting from randomly shuffled labels. This dataset is likewise fully separable when projected to the first $P$ components, but its labels are uncorrelated with the original labels (for $P = 500$: $p = 0.06$, $r < 10^{-10}$).

The mean training accuracy of 10 non-linear networks with $P$=10,50,500 is plotted in Fig. 9a (first dataset) and Fig. 9b (second dataset). In both cases, the lower $P$ is (namely, only the first few principal components are discriminative), the faster the data is learned by the non-linear network. Whether the labels are real or shuffled makes little qualitative difference, as predicted by the *PC-bias*.

## 6   Summary and discussion

When trained with gradient descent, the convergence rate of the over-parameterized deep linear network model is provably governed by the eigendecomposition of the data, and specifically, parameters corresponding to the most significant principal components converge faster than the least significant components. Empirical evidence is provided for the relevance of these results to more realistic non-linear networks. We term this effect *PC-bias*. This result provides a complementary account for some prevalent empirical observations, including the benefit of early stopping and the slower convergence rate with shuffled labels.

We use the *PC-bias* to explicate the *Learning Order Constancy (LOC)*, showing that examples learned at earlier stages are more distinguishable by the higher principal components, demonstrating that networks' training relies more heavily on higher principal components early on. A causal link between the *PC-bias* and the *LOC-effect* is demonstrated, as the *LOC-effect* diminishes when the *PC-bias* is eliminated by whitening the images. We analyze these findings in view of a related phenomenon termed *spectral bias*. While the *PC-bias* may be more prominent early on, the *spectral bias* may be more important in later stages of learning.

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
