**Supplementary Material**

 **A   Random Matrices**

 **A.1   Multiplication of Random Matrices**

477 In this section we present and prove some statistical properties of general random matrices and their
478 multiplications. Let $\{Q_n \in \mathbb{R}^{m_n \times m_{n-1}}\}_{n=1}^N$ denote a set of random matrix whose elements are
479 sampled iid from a distribution with mean $0$ and variance $\sigma_n^2$, with bounded kurtosis. Let

$$\boldsymbol{Q}^l = \prod_{n=l-1}^1 Q_n = Q_{l-1} \cdot \ldots \cdot Q_1, \qquad B^l = \boldsymbol{Q}^{l\top}\boldsymbol{Q}^l \in \mathbb{R}^{m_0 \times m_0} \quad l \in [2 \ldots N]$$

$$\mathsf{Q}^l = \prod_{n=N}^{l+1} Q_n = Q_N \cdot \ldots \cdot Q_{l+1}, \qquad A^l = \mathsf{Q}^l\mathsf{Q}^{l\top} \in \mathbb{R}^{m_N \times m_N} \quad l \in [1 \ldots N-1]$$

(6)

480 **Theorem 3.** $\forall l$

$$\mathbb{E}(B^l) = \beta_l I \qquad \beta_l = \prod_{n=1}^{l-1} m_n \sigma_n^2 \tag{7}$$

$$\mathbb{E}(A^l) = \alpha_l I \qquad \alpha_l = \prod_{n=l+1}^N m_{n-1}\sigma_n^2 \tag{8}$$

481 *Proof.* We only prove (7), as the proof of (8) is similar. To simplify the presentation, we use the
482 following auxiliary notations: $V = Q_1, U = \prod_{n=l-1}^2 Q_n \implies \boldsymbol{Q}^l = UV$.

483 Proof proceeds by induction on $l$.

484     • $l = 2$:

$$\mathbb{E}[B_{ij}^l] = \mathbb{E}[\sum_{k=1}^{m_1} V_{ki}V_{kj}] \overset{i \neq j}{=} \sum_{k=1}^{m_1} \mathbb{E}[V_{ki}]\mathbb{E}[V_{kj}]$$

$$\mathbb{E}[B_{ii}^l] = \mathbb{E}[\sum_{k=1}^{m_1} V_{ki}V_{ki}] = \sum_{k=1}^{m_1} \mathbb{E}[V_{ki}^2]$$

485     Thus

$$\mathbb{E}[B_{ij}^l] = \begin{cases} 0 & i \neq j \quad \text{(off diagonal)} \\ m_1\sigma_1^2 & i = j \quad \text{(diagonal)} \end{cases}$$

486     • Assume that (7) holds for $l - 1$.

$$B_{ij}^l = \sum_k \boldsymbol{Q}_{ki}^l \boldsymbol{Q}_{kj}^l = \sum_k \sum_\nu U_{k\nu}V_{\nu i} \sum_\rho U_{k\rho}V_{\rho j}$$

487     and therefore

$$\mathbb{E}[B_{ij}^l] = \sum_k \sum_\nu \sum_\rho \mathbb{E}[U_{k\nu}V_{\nu i}U_{k\rho}V_{\rho j}] = \sum_\nu \sum_\rho \mathbb{E}[V_{\nu i}V_{\rho j}] \sum_k \mathbb{E}[U_{k\nu}U_{k\rho}]$$

488     where the last transition follows from the independence of $U$ and $V$. Once again, we
489     consider the diagonal and off-diagonal elements separately. If $i \neq j$:

$$\mathbb{E}[B_{ij}^l] = \sum_\nu \sum_\rho \mathbb{E}[V_{\nu i}]\mathbb{E}[V_{\rho j}] \sum_k \mathbb{E}[U_{k\nu}U_{k\rho}] = 0$$

490     If $i = j$:

$$\mathbb{E}[B_{ii}^l] = \sum_\nu \sum_\rho \mathbb{E}[V_{\nu i}V_{\rho j}]\mathbb{E}[(U^\top U)_{\nu\rho}] = \sum_\nu \mathbb{E}[V_{\nu i}^2]\mathbb{E}[(U^\top U)_{\nu\nu}]$$

491  Using the induction assumption

$$\mathbb{E}[B_{ij}^l] = \begin{cases} 0 & i \neq j \quad \text{(off diagonal)} \\ m_1\sigma_1^2 \prod_{n=2}^{l-1} m_n\sigma_n^2 & i = j \quad \text{(diagonal)} \end{cases}$$

492  from which (7) follows.  □

493  Let $\mathrm{m}$ denote the width of the smallest hidden layer, $\mathrm{m} = \min(m_1, \ldots, m_{N-1})$, and assume that
494  $\max(m_1, \ldots, m_{N-1}) - \min(m_1, \ldots, m_{N-1})$ is bounded by some $M_b$ as $\mathrm{m} \to \infty$. Assume the
495  following initialization scheme

496  **Definition 4.** *The elements of $\{Q_n\}_{n=1}^N$ are chosen iid from a distribution with mean 0 and variance*
497  $\sigma_n^2$, *where*

$$\sigma_n^2 = \frac{2}{m_{n-1} + m_n} \quad 1 < n < N, \qquad \sigma_1^2 = \frac{1}{m_1}, \qquad \sigma_N^2 = \frac{1}{m_{N-1}}$$

498  For large $\mathrm{m}$, it follows that

$$m_n\sigma_n^2 = 1 + O\left(\frac{1}{\mathrm{m}}\right) \qquad n \in [1 \ldots N - 1]$$

$$m_{n-1}\sigma_n^2 = 1 + O\left(\frac{1}{\mathrm{m}}\right) \qquad n \in [2 \ldots N]$$

499  **Corollary 3.1.** *With initialization as in Def. 4, $\forall l$*

$$\mathbb{E}(B^l) = [1 + O\left(\frac{1}{\mathrm{m}}\right)]I, \qquad \mathbb{E}(A^l) = [1 + O\left(\frac{1}{\mathrm{m}}\right)]I$$

500  **Theorem 4.** *With initialization as in Def. 4, $\forall l$*

$$\mathrm{var}(B^l) = O\left(\frac{1}{\mathrm{m}}\right), \qquad \mathrm{var}(A^l) = O\left(\frac{1}{\mathrm{m}}\right)$$

501  *Proof.* We prove by induction on $l$ that:

$$\mathbb{E}[(B_{ij}^l)^2] = \begin{cases} O\left(\frac{1}{\mathrm{m}}\right) & i \neq j \quad \text{(off diagonal)} \\ 1 + O\left(\frac{1}{\mathrm{m}}\right) & i = j \quad \text{(diagonal)} \end{cases}, \qquad \mathbb{E}[B_{ii}^l B_{jj}^l] = 1 + O\left(\frac{1}{\mathrm{m}}\right) \qquad (9)$$

502  For $l = 2$, (9) follows from Lemma 2 and Corr 3.1. We now assume that (9) holds for $l - 1$ and prove
503  for $l$, using notations as above: $V = Q_1$, $U = \prod_{l-1}^2 Q_n$, $\mathbf{Q}^l = UV$.

$$\mathbb{E}[(B_{ij}^l)^2] = \sum_{\nu,\rho} \sum_{\alpha,\beta} \mathbb{E}[V_{\nu i}V_{\rho j}V_{\alpha i}V_{\beta j} \sum_{k,n} U_{k\nu}U_{k\rho}U_{n\alpha}U_{n\beta}]$$

504  Let $B' = U^\top U$. Using the induction assumption

$$\mathbb{E}[(B_{ij}^l)^2] \stackrel{i \neq j}{=} \sum_{\nu,\rho} \mathbb{E}[V_{\nu i}^2 V_{\nu j}^2]\mathbb{E}[(B'_{\nu\rho})^2] = O\left(\frac{1}{\mathrm{m}}\right)$$

505  When $i = j$, there are 3 cases where the terms in the sum above do not equal 0: (i) $\nu = \alpha$, $\rho =$
506  $\beta, \nu \neq \rho$ or $\nu = \beta$, $\rho = \alpha, \nu \neq \rho$; (ii) $\nu = \rho$, $\alpha = \beta, \nu \neq \alpha$; (iii) $\nu = \rho = \alpha = \beta$. Case (i) is
507  similar to the above, and we therefore only expand cases (ii) and (iii) next:

$$\text{(ii)} \sum_{\nu,\alpha} \mathbb{E}[V_{\nu i}^2 V_{\alpha i}^2]\mathbb{E}[B'_{\nu\nu}B'_{\alpha\alpha}] = 1 + O\left(\frac{1}{\mathrm{m}}\right)$$

$$\text{(iii)} \sum_{\nu} \mathbb{E}[V_{\nu i}^4]\mathbb{E}[(B'_{\nu\nu})^2] = O\left(\frac{1}{\mathrm{m}}\right)$$

508  In the derivation of (iii) we exploit the assumption that the kurtosis of the distribution used to sample
509  $Q_n$ is fixed at $G$ and cannot depend on $\mathrm{m}$, indicating that $\mathbb{E}[V_{\nu i}^4] = G\sigma_1^4$.

510  A similar argument would show that $\mathbb{E}[B_{ii}^l B_{jj}^l] = 1 + O\left(\frac{1}{\mathrm{m}}\right)$.  □

**Theorem 5.** *Let $\{X(\mathrm{m})\}$ denote a sequence of random matrices where $\mathbb{E}[X(\mathrm{m})] = [1 + O(\frac{1}{\mathrm{m}})]I$ and $\mathrm{var}[X(\mathrm{m})] = O(\frac{1}{\mathrm{m}})$. Then $X(\mathrm{m}) \xrightarrow{p} I$, where $\xrightarrow{p}$ denotes convergence in probability.*

*Proof.* We need to show that $\forall \epsilon, \delta > 0 \; \exists \mathrm{m}' \in \mathbb{N}$, such that $\forall \mathrm{m} > \mathrm{m}'$

$$P\left(|X(m) - I| > \varepsilon\right) < \delta$$

Henceforth we use $X$ as shorthand for $X(\mathrm{m})$. Since $\mathbb{E}(X) = [1 + O(\frac{1}{\mathrm{m}})]I$, it follows that $\forall \varepsilon > 0$ $\exists \mathrm{m}_1 \in \mathbb{N}$ such that $\forall \mathrm{m} > \mathrm{m}_1$, the following holds element-wise:

$$|\mathbb{E}(X) - I| < \frac{\varepsilon}{2}$$

Thus

$$P\left(|X - I| > \varepsilon\right) \leq P\left(|X - \mathbb{E}(X)| > \frac{\varepsilon}{2}\right)$$

Since $\mathrm{var}(X) = O(\frac{1}{\mathrm{m}})$, it follows that $\forall \varepsilon, \delta > 0, \; \exists \mathrm{m}_2 \in \mathbb{N} \; \ni \; \forall \mathrm{m} > \mathrm{m}_2$

$$\mathrm{var}(X) < \frac{\varepsilon^2}{4}\delta$$

From the above, and using Chebyshev inequality

$$P\left(|X - I| > \varepsilon\right) < \frac{4\mathrm{var}(X)}{\varepsilon^2} < \delta$$

$\forall \mathrm{m} > \mathrm{m}'$, where $\mathrm{m}' = \max\{\mathrm{m}_1, \mathrm{m}_2\}$.

$\square$

Let $A^l(m)$ and $B^l(\mathrm{m})$ denote a sequence of random matrices as defined in (6), corresponding to models for which $m = \min(m_1, ..., m_{L-1})$.

**Corollary 5.1.**

$$B^l(m) \xrightarrow{p} I \quad \forall l \in [2 \dots N] \qquad A^l(m) \xrightarrow{p} I \quad \forall l \in [1 \dots N-1]$$

The proof follows from Corr 3.1, Thm 4 and Thm 5.

## A.2 Dynamics of Random Matrices

Consider a dynamical process, where the random matrices defined above are changed as $Q_j \rightarrow Q_j - \Delta Q_j \; \forall j$, and specifically from (21):

$$\Delta Q_j = \mu \Big(\prod_{n=N}^{j+1} Q_n\Big)^\top E_r \Big(\prod_{n=j-1}^{1} Q_n\Big)^\top, \qquad E_r = \Big(\prod_{n=N}^{1} Q_n\Big)\Sigma_{XX} - \Sigma_{YX} \tag{10}$$

Denoting $\boldsymbol{Q}^l \rightarrow \boldsymbol{Q}^l - \Delta \boldsymbol{Q}^l$ and applying the product rule

$$\Delta \boldsymbol{Q}^l = \sum_{j=1}^{l-1} \Big(\prod_{n=l-1}^{j+1} Q_n\Big) \Delta Q_j \Big(\prod_{n=j-1}^{1} Q_n\Big) \tag{11}$$

For $B^l = {\boldsymbol{Q}^l}^\top \boldsymbol{Q}^l$ and denoting $B^l \rightarrow B^l - \Delta B^l$:

$$\Delta B^l = [\Delta {\boldsymbol{Q}^l}^\top \boldsymbol{Q}^l + {\boldsymbol{Q}^l}^\top \Delta \boldsymbol{Q}^l] \tag{12}$$

Before proceeding to analyze $\Delta B^l$, we note that

$$m_N \sigma_N^2 = \frac{K}{m_{N-1}} = \frac{K}{\mathrm{m}}[1 + O\left(\frac{1}{\mathrm{m}}\right)]$$

and therefore from Thm 3

$$\mathbb{E}[({Q^N}^\top)Q^N] = [\frac{K}{\mathrm{m}} + O\left(\frac{1}{\mathrm{m}}\right)]I \tag{13}$$

**Theorem 6.** *For sequence $B^l(m)$ defined as above, if*

$$B^l(m) \xrightarrow{p} I, \quad \mathrm{var}[B^l(m)] = O\Big(\frac{1}{\mathrm{m}}\Big)$$

*then*

$$\Delta B^l(m) \xrightarrow{p} 0, \quad \mathrm{var}[\Delta B^l(m)] = O\Big(\frac{1}{\mathrm{m}}\Big)$$

*Proof.* $B^l(m) \xrightarrow{p} I$ implies that $\forall \epsilon, \delta > 0 \; \exists \hat{\mathrm{m}} \in \mathbb{N}$, such that $\forall \mathrm{m} > \hat{\mathrm{m}}$ and with probability larger than $1 - \delta$.

$$B^l(m) = I + e_1 \quad |e_1| < \varepsilon, \; \forall l \in [2 \dots N] \tag{14}$$

In addition, from (13) and Thms. 4-5

$$Q^{N^\top} Q^N = \frac{K}{\mathrm{m}} I + e_2 \quad |e_2| < \varepsilon$$

We fix $\mathrm{m}$ and let $B^l$ be a shorthand for $B^l(m)$. Now

$$B^{N+1} = \boldsymbol{Q}_{N-1}{}^\top Q^{N^\top} Q^N \boldsymbol{Q}_{N-1} = \frac{K}{\mathrm{m}} \boldsymbol{Q}^{N-1^\top} \boldsymbol{Q}^{N-1} + O(\varepsilon) = \frac{K}{\mathrm{m}} I + O(\varepsilon) \tag{15}$$

To evaluate $\Delta B^l$ from (12), we start from

$$\boldsymbol{Q}^{l^\top} \Delta \boldsymbol{Q}^l = \sum_{j=1}^{l-1} \Big(\prod_{l-1}^{1} Q_n\Big)^\top \Big(\prod_{l-1}^{j+1} Q_n\Big) \Delta Q_j \prod_{j-1}^{1} Q_n$$

Simplifying $t_j$ – the $j^{\text{th}}$ term in the sum

$$t_j = \mu \prod_{1}^{l-1} Q_n^\top \prod_{l-1}^{j+1} Q_n \prod_{j+1}^{N} Q_n^\top E_r \prod_{1}^{j-1} Q_n^\top \prod_{j-1}^{1} Q_n = \mu B^l (B^{j+1})^{-1} \boldsymbol{Q}^{N^\top} E_r B^j + O(\varepsilon)$$

The last transition is exactly true when $B^l = I$ and $B^{j+1} = I$, as shown in Lemma 3 in §A.3. Substituting $E_r$

$$\begin{aligned}
t_j &= \mu B^l (B^{j+1})^{-1} \boldsymbol{Q}^{N^\top} [\boldsymbol{Q}^N \Sigma_{XX} - \Sigma_{YX}] B^j + O(\varepsilon) \\
&= \mu B^l (B^{j+1})^{-1} [B^{N+1} \Sigma_{XX} + \boldsymbol{Q}^{N^\top} \Sigma_{YX}] B^j + O(\varepsilon)
\end{aligned}$$

Substituting (14) and (15)

$$t_j = \mu [\frac{K}{\mathrm{m}} \Sigma_{XX} + \boldsymbol{Q}^{N^\top} \Sigma_{YX}] + O(\varepsilon) \tag{16}$$

From (16) and Lemma 2

$$\mathbb{E}[\boldsymbol{Q}^{l^\top} \Delta \boldsymbol{Q}^l] = \sum_{j=1}^{l-1} \mathbb{E}[t_j] = \mu l \frac{K}{\mathrm{m}} I + O(\varepsilon)$$

Since $\Delta \boldsymbol{Q}^{l^\top} \boldsymbol{Q}^l = [\boldsymbol{Q}^{l^\top} \Delta \boldsymbol{Q}^l]^\top$, it follows from (12) that

$$\mathbb{E}[\Delta B^l] = 2\mu l \frac{K}{\mathrm{m}} I + O(\varepsilon) \tag{17}$$

To conclude the proof, we need to show that $\forall \epsilon', \delta' > 0 \; \exists \hat{\mathrm{m}}' \in \mathbb{N}$, such that $\forall \mathrm{m} > \hat{\mathrm{m}}'$

$$P\left(|\Delta B^l| > \varepsilon'\right) < \delta'$$

Since (17) is true with probability $(1 - \delta) \; \forall \varepsilon, \delta$ and $\forall \mathrm{m} > \hat{\mathrm{m}}$, we choose $\varepsilon$ and $\hat{\mathrm{m}}'$ such that

$$|\mathbb{E}[\Delta B^l]| < \frac{\varepsilon'}{2} \quad \forall \mathrm{m} > \hat{\mathrm{m}}' \tag{18}$$

$$P\left(|\Delta B^l| > \varepsilon'\right) \leq (1-\delta)P\left(|\Delta B^l - \mathbb{E}(\Delta B^l)| > \frac{\varepsilon'}{2}\right) < \frac{4\mathrm{var}(\Delta B^l)}{\varepsilon'^2}(1-\delta)$$

$\mathrm{var}(\Delta B^l) = O(\frac{1}{\mathrm{m}})$ implies that $\exists \hat{\mathrm{m}}'' \in \mathbb{N}, \delta > 0$, such that $\forall \mathrm{m} > \hat{\mathrm{m}}''$

$$\frac{4\mathrm{var}(\Delta B^l)}{\varepsilon'^2}(1-\delta) < \delta'$$

It now follows that $\Delta B^l(m) \xrightarrow{p} 0$.

To analyze the variance, we assume that all the moments of the distribution functions used to sample $Q_n$ are bounded. Thus, from (16), the variance of $t_j \,\forall j$ remains $O(\frac{1}{\mathrm{m}})$. Likewise, since $\Delta B^l$ is a sum of matrices, each with variance $O(\frac{1}{\mathrm{m}})$ thus bounding the covariance by $O(\frac{1}{\mathrm{m}})$, we can deduce that $\mathrm{var}(\Delta B^l) = O(\frac{1}{\mathrm{m}})$.

$\square$

**Theorem 7.** *For sequence $A^l(m)$ defined as above, if*

$$A^l(m) \xrightarrow{p} I \quad \text{and} \quad \mathrm{var}[A^l(m)] = O\left(\frac{1}{\mathrm{m}}\right)$$

*then*

$$\Delta A^l(m) \xrightarrow{p} 0 \quad \text{and} \quad \mathrm{var}[\Delta A^l(m)] = O\left(\frac{1}{\mathrm{m}}\right)$$

The proof is mostly similar to Thm 6, though we additionally need to show the following in order to replace (13):

$$\mathbb{E}[\mathbf{Q}^0 \Sigma_{XX} \mathbf{Q}^{0\top}] = [\frac{q}{\mathrm{m}} + O\left(\frac{1}{\mathrm{m}}\right)]I$$

This, in turn, can be proved in a similar manner to the proof of Thm 3, when taking into account the initialization scheme defined in Def. 4.

**Note about convergence rate.** In Thm 6, convergence to 0 when $\mathrm{m} \to \infty$ is governed by $O\left(\frac{K}{\mathrm{m}}\right)$. In Thm 7, convergence is governed by $O(\frac{q}{\mathrm{m}})$.

## A.3 Some Useful Lemmas

**Lemma 1.** *Given function $G(W) = \frac{1}{2}\|UWVX - Y\|_F^2$, its derivative is the following*

$$\frac{dG(W)}{dW} = U^\top UWVX(VX)^\top - U^\top Y(VX)^\top = U^\top[UWV\Sigma_{XX} - \Sigma_{YX}]V^\top \qquad (19)$$

**Lemma 2.** *Given $\mathbf{Q} = \prod_{n=N}^1 Q_n$, where $Q_n \in \mathbb{R}^{m_n \times m_{n-1}}$ denotes a random matrix whose elements are sampled iid from a distribution with mean 0 and variance $\sigma_n^2$, $\forall i, j$.*

$$\mathbb{E}[\mathbf{Q}_{ij}] = 0 \qquad \mathrm{var}[\mathbf{Q}_{ij}] = \frac{1}{m_N}\prod_{n=1}^N m_n \sigma_n^2 \qquad (20)$$

*Proof.* By induction on $N$. Clearly for $N = 1$:
$$\mathbb{E}[\mathbf{Q}_{ij}] = \mathbb{E}[(Q_1)_{ij}] = 0 \qquad \mathrm{var}[\mathbf{Q}_{ij}] = \mathrm{var}[(Q_1)_{ij}] = \sigma_1^2$$

Assume that (20) holds for $N - 1$. Let $V = \prod_{n=N-1}^1 Q_n$, $U = Q_N$. It follows that
$$\mathbb{E}[\mathbf{Q}_{ij}] = \mathbb{E}[(UV)_{ij}] = \sum_k \mathbb{E}[U_{ik}V_{kj}] = \sum_k \mathbb{E}[U_{ik}]\mathbb{E}[V_{kj}] = 0$$

where the last transition follows from the independence of $U$ and $V$. In a similar manner
$$\mathrm{var}[\mathbf{Q}_{ij}] = \mathbb{E}[\mathbf{Q}_{ij}^2] = \mathbb{E}[(\sum_k U_{ik}V_{kj})^2] = \mathbb{E}[\sum_k U_{ik}V_{kj}\sum_l U_{il}V_{lj}] = \sum_k \mathbb{E}[U_{ik}^2]\mathbb{E}[V_{kj}]^2$$

$$= m_{N-1}\sigma_N^2 \frac{1}{m_{N-1}}\prod_{n=1}^{N-1} m_n \cdot \sigma_n^2 = \frac{1}{m_N}\prod_{n=1}^N m_n \cdot \sigma_n^2$$

569 With the initialization scheme defined in Def. 4, $\text{var}(\boldsymbol{Q}_{ij}) = O(\frac{1}{m})$. □

570 **Lemma 3.** *Consider matrix multiplication $CD$ where $C \in \mathbb{R}^{k \times m}$, $D \in \mathbb{R}^{m \times k}$, $k \ll m$ and*
571 $\text{rank}(CD) = k$. *Define $\Delta_1 \mathbb{R}^{m \times k}$, $\Delta_2 \mathbb{R}^{k \times m}$. Then*

$$C\Delta_1 = \Delta_2\Delta_1 = I \implies CD = C\Delta_1\Delta_2 D$$

572 *Proof.* Since $C = \Delta_1^+$ and $\Delta_2 = \Delta_1^+$

$$C = \Delta_1^+\Delta_1 C = C\Delta_1\Delta_1^+ = C\Delta_1\Delta_2$$

573 □

# B  Supplementary Proofs and Additional Models

## B.1  Deep Linear networks

576 Here we prove Thm 1 as defined in Section 2.1.

577 **Theorem 1.** *The compact matrix representation $\boldsymbol{W}$ obeys the following dynamics*

$$\boldsymbol{W}^{s+1} = \boldsymbol{W}^s - \mu \sum_{l=1}^{L} A_l^s \cdot Er^s \cdot B_l^s + O(\mu^2)$$

578 *where the gradient scale matrices $A_l^s$, $B_l^s$ are defined in (3)*

$$A_l^s := \left( \prod_{j=L}^{l+1} W_j^s \right)\left( \prod_{j=L}^{l+1} W_j^s \right)^{\top} \in \mathbb{R}^{K \times K} \qquad B_l^s := \left( \prod_{j=l-1}^{1} W_j^s \right)^{\top}\left( \prod_{j=l-1}^{1} W_j^s \right) \in \mathbb{R}^{q \times q}$$

579

580 *Proof.* At time $s$, the gradient step $\Delta W_l^s$ of layer $l$ is defined by differentiating $L(\mathbb{X})$ with respect to
581 $W_l^s$. Henceforth we omit index $s$ for clarity. First, we rewrite $L(\mathbb{X})$ as follows:

$$L(\mathbb{X}; W_l) = \frac{1}{2}\| \left( \prod_{j=L}^{l+1} W_j \right) W_l \left( \prod_{j=l-1}^{1} W_j \right) X - Y\|_F^2$$

582 Differentiating $L(\mathbb{X}; W_l)$ to obtain the gradient $\Delta W_l = \frac{\partial L(\mathbb{X}; W_l)}{\partial W_l}$, using Lemma 1 above, we get

$$\Delta W_l = \left( \prod_{j=L}^{l+1} W_j \right)^{\top} [\boldsymbol{W}\Sigma_{XX} - \Sigma_{YX}]\left( \prod_{l-1}^{1} W_j \right)^{\top} \tag{21}$$

583 Finally

$$\Delta \boldsymbol{W} = \prod_{l=L}^{1} (W_l - \mu\Delta W_l) - \prod_{l=L}^{1} W_l = -\mu \sum_{l=1}^{L} \left( \prod_{n=L}^{l+1} W_n \right) \Delta W_l \left( \prod_{n=l-1}^{1} W_n \right) + O(\mu^2)$$

584 Substituting $\Delta W_l$ and $Er$ (as defined in Def. 3) into the above completes the proof.

585 □

## B.2  Adding Non-Linear ReLU Activation

587 The results shown in Fig. 2b pertain to a relatively simple non-linear model analyzed by Arora et al.
588 (2019), here adapted to classification rather than regression. Specifically, it is a two-layer model with
589 ReLU activation, where only the weights of the first layer are being learned. Similarly to (1), the loss
590 is defined as

$$L(\mathbb{X}) = \frac{1}{2}\sum_{i=1}^{n} \|f(\boldsymbol{x}_i) - \boldsymbol{y}_i\|^2 \qquad f(\boldsymbol{x}_i) = \boldsymbol{a}^{\intercal} \cdot \sigma(W\boldsymbol{x}_i), \ \boldsymbol{a} \in \mathbb{R}^m, \ W \in \mathbb{R}^{m \times d}$$

591 m denotes the number of neurons in the hidden layer. We consider a binary classification problem
592 with 2 classes, where $y_i = 1$ for $\boldsymbol{x}_i \in C_1$, and $y_i = -1$ for $\boldsymbol{x}_i \in C_2$. $\sigma(.)$ denotes the ReLU
593 activation function applied element-wise to vectors, where $\sigma(u) = u$ if $u \geq 0$, and 0 otherwise.

594 At time $s$, each gradient step is defined by differentiating $L(\mathbb{X})$ with respect to $W$. Due to the
595 non-linear nature of the activation function $\sigma(.)$, we separately[3] differentiate each row of $W$, denoted
596 $\boldsymbol{w}_r$ where $r \in [\mathrm{m}]$, as follows:

$$\boldsymbol{w}_r^{s+1} - \boldsymbol{w}_r^s = -\mu \frac{\partial L(\mathbb{X})}{\partial \boldsymbol{w}_r}\Big|_{\boldsymbol{w}_r = \boldsymbol{w}_r^s} = -\mu \sum_{i=1}^n \Big[ \boldsymbol{a}^\intercal \cdot \sigma(W^s \boldsymbol{x}_i) - y_i \Big] \frac{\partial f(\boldsymbol{x}_i)}{\partial \boldsymbol{w}_r}\Big|_{\boldsymbol{w}_r = \boldsymbol{w}_r^s}$$

$$= -\mu \sum_{i=1}^n \Big[ \sum_{j=1}^{\mathrm{m}} a_j \sigma(\boldsymbol{w}_j^s \cdot \boldsymbol{x}_i) - y_i \Big] a_r \boldsymbol{x}_i^\intercal \mathbb{1}_{\boldsymbol{w}_r^s}(\boldsymbol{x}_i)$$

$$= -\mu a_r \sum_{i=1}^n \mathbb{1}_{\boldsymbol{w}_r^s}(\boldsymbol{x}_i) \Big[ \Psi^s(\boldsymbol{x}_i) \cdot \boldsymbol{x}_i - y_i \Big] \boldsymbol{x}_i^\intercal \qquad \text{where } \Psi^s(\boldsymbol{x}_i) = \sum_{j=1}^{\mathrm{m}} a_j \boldsymbol{w}_j^s \mathbb{1}_{\boldsymbol{w}_j^s}(\boldsymbol{x}_i)$$

597 Above $\mathbb{1}_{\boldsymbol{w}_r^s}(\boldsymbol{x}_i)$ denotes the indicator function that equals 1 when $\boldsymbol{w}_r^s \cdot \boldsymbol{x}_i \geq 0$, and 0 otherwise.

598 In order to proceed, we make two assumptions:

    599     1. The distribution of the data is symmetric where $P(\boldsymbol{x}_i) = P(-\boldsymbol{x}_i)$.

    600     2. $W$ and $\boldsymbol{a}$ are initialized so that $\boldsymbol{w}_{2i}^0 = -\boldsymbol{w}_{2i-1}^0$ and $a_{2i} = -a_{2i-1}$ $\forall i \in [\frac{\mathrm{m}}{2}]$.

601 It follows from Assumption 2 that at the beginning of training $\mathbb{1}_{\boldsymbol{w}_{2j}^0}(\boldsymbol{x}_i) + \mathbb{1}_{\boldsymbol{w}_{2j-1}^0}(\boldsymbol{x}_i) = 1, \forall \boldsymbol{x}_i$
602 such that $\boldsymbol{w}_{2j-1}\boldsymbol{x}_i \neq \boldsymbol{w}_{2j}\boldsymbol{x}_i \neq 0$, and $\forall j \in [\frac{\mathrm{m}}{2}]$. Consequently

$$\Psi^0(\boldsymbol{x}_i) = \sum_{j=1}^{\mathrm{m}} a_j \boldsymbol{w}_j^0 \mathbb{1}_{\boldsymbol{w}_j^s}(\boldsymbol{x}_i) = \frac{1}{2} \sum_{j=1}^{\mathrm{m}} a_j \boldsymbol{w}_j^0 = \frac{1}{2} \boldsymbol{a}^\intercal W^0$$

603 $\forall \boldsymbol{x}_i$ such that $\boldsymbol{w}_{2j-1}\boldsymbol{x}_i \neq \boldsymbol{w}_{2j}\boldsymbol{x}_i \neq 0$. Finally

$$\boldsymbol{w}_r^1 - \boldsymbol{w}_r^0 = -\mu a_r \Big[ \frac{1}{2} \boldsymbol{a}^\intercal W^0 \sum_{\substack{i=1 \\ \boldsymbol{w}_r^0 \boldsymbol{x}_i \geq 0}}^n \boldsymbol{x}_i \boldsymbol{x}_i^\intercal - \sum_{\substack{i=1 \\ \boldsymbol{w}_r^0 \boldsymbol{x}_i \geq 0}}^n y_i \boldsymbol{x}_i^\intercal \Big]$$

604 Next, we note that Assumption 1 implies

$$\mathbb{E}\Big[ \sum_{\substack{i=1 \\ \boldsymbol{w} \cdot \boldsymbol{x}_i \geq 0}}^n \boldsymbol{x}_i \boldsymbol{x}_i^\intercal \Big] = \frac{1}{2} \mathbb{E}\Big[ \sum_{i=1}^n \boldsymbol{x}_i \boldsymbol{x}_i^\intercal \Big] = \frac{1}{2} \mathbb{E}[\Sigma_{XX}]$$

605 for any vector $\boldsymbol{w}$. Thus, if the sample-size $n$ is large enough, at the beginning of training we expect
606 to see

$$\boldsymbol{w}_r^{s+1} - \boldsymbol{w}_r^s \approx -\mu \frac{a_r}{2} [\boldsymbol{a}^\intercal W^s \Sigma_{XX} - \tilde{\boldsymbol{m}}_r^s] \quad \forall r$$

607 where row vector $\tilde{\boldsymbol{m}}_r^s$ denotes the vector difference between the centroids of classes $C_1$ and $C_2$,
608 computed in the half-space defined by $\boldsymbol{w}_r^s \cdot \boldsymbol{x} \geq 0$. Finally (for small $s$)

$$W^{s+1} - W^s \approx -\mu \frac{1}{2} \Big[ (\boldsymbol{a}\boldsymbol{a}^\intercal) W^s \Sigma_{XX} - \tilde{M}^s \Big]$$

609 where $\tilde{M}^s$ denotes the matrix whose $r$-th row is $a_r \tilde{\boldsymbol{m}}_r^s$. This equation is reminiscent of the single
610 layer linear model dynamics $\boldsymbol{W}^{s+1} = \boldsymbol{W}^s - \mu E r^s$, and we may conclude that when it holds and
611 using the principal coordinate system, the rate of convergence of the $j-$th column of $W^s$ is governed
612 by the singular value $d_j$ .

---

[3]Since the ReLU function is not everywhere differentiable, the following may be considered the definition of
the update rule.

### B.3 Weight Evolution

To analyze the weight dynamics, we first shift to the principal coordinate system defined in Def 1. In this representation $Er^s = W^s D - M$, where $D = diag(\{d_j\}_{j=1}^q)$ is a diagonal matrix. Based on Thm 1 and the subsequent discussion of convergence rate, assuming that the width of the hidden layers is very large, we can readily substitute $B_l^s \approx I \ \forall l$ in (2), to obtain

$$\boldsymbol{W}^{s+1} = \boldsymbol{W}^s - \mu \sum_{l=1}^{L} A_l^s Er^s + O(\mu^2) \qquad (22)$$

Let $\boldsymbol{w}_j \in \mathbb{R}^K$ denote the $j$-th column of $\boldsymbol{W}$, $\boldsymbol{m}_j$ denote the $j$-th column of $M$. From (22) we have

$$\boldsymbol{w}_j^{s+1} = \boldsymbol{w}_j^s - \mu \sum_{l=1}^{L} A_l^s (d_j \boldsymbol{w}_j^s - \boldsymbol{m}_j) \quad j \in [K]$$

This is a telescoping series; denoting $A^s = \sum_{l=1}^{L} A_l^s$,

$$
\begin{aligned}
\boldsymbol{w}_j^{s+1} =& \boldsymbol{w}_j^s - \mu A^s (d_j \boldsymbol{w}_j^s - \boldsymbol{m}_j) = (I - \mu d_j A^s) \boldsymbol{w}_j^s + \mu A^s \boldsymbol{m}_j = \ldots \\
=& \prod_{\nu=1}^{s} (I - \mu d_j A^\nu) \boldsymbol{w}_j^0 + \mu \left[ \sum_{\nu=1}^{s} \prod_{\rho=\nu+1}^{s} (I - \mu d_j A^\rho) A^\nu \right] \boldsymbol{m}_j
\end{aligned}
\qquad (23)
$$

The only difference between individual columns lies in $d_j$, which governs the rate of convergence of the first term to 0, and the rate of convergence of the second term to the optimal value of $\frac{1}{d_j} \boldsymbol{m}_j$.

In the discussion following the proof of Thm 2, we noted that the approximation $A_l^s \approx I$ breaks down before $B_l^s \approx I$. Nevertheless, while it is still valid, (23) further simplifies to the following

$$
\begin{aligned}
\boldsymbol{w}_j^{s+1} &= (1 - \mu d_j L)^s \boldsymbol{w}_j^0 + \mu \left[ \sum_{\nu=1}^{s} (1 - \mu d_j L)^{(s-\nu)} L I \right] \boldsymbol{m}_j = \lambda_j^s \boldsymbol{w}_j^0 + \mu L \left[ \sum_{k=0}^{s-1} \lambda_j^k \right] \boldsymbol{m}_j \\
&= \lambda_j^s \boldsymbol{w}_j^0 + (1 - \lambda_j^s) \frac{\boldsymbol{m}_j}{d_j} \qquad\qquad \lambda_j = 1 - \mu d_j L
\end{aligned}
$$

## C  Additional Empirical Results

### C.1  Weight Initialization

We evaluate empirically the weight initialization scheme from Def. 4. When compared to Glorot uniform initialization (Glorot & Bengio, 2010), the only difference between the two schemes lies in how the first and last layers are scaled. Thus, in order to highlight the difference between the methods, we analyze a fully connected linear network with a single hidden layer, whose dimension (the number of hidden neurons) is much larger than the input and output dimensions. We trained $N$=10 such networks on a binary classification problem, once with the initialization suggested in Def. 4, and again with Glorot uniform initialization. While both initialization schemes achieve the same final accuracy upon convergence, our proposed initialization variant converges faster on both train and test datasets (see Fig. 10).

### C.2  Spectral Bias

The *spectral bias*, discussed in Section 4.3, can also induce similar learning order in different networks. To support the discussion in Section 4.3, in §C.2.2) we analyze the relation between the *spectral bias* and *accessibility*, in order to clarify its relation to the *Learning Order Constancy* and the *PC-bias*. First, however, we expand the scope of the empirical evidence for this effect to the classification scenario and real image data (§C.2.1).

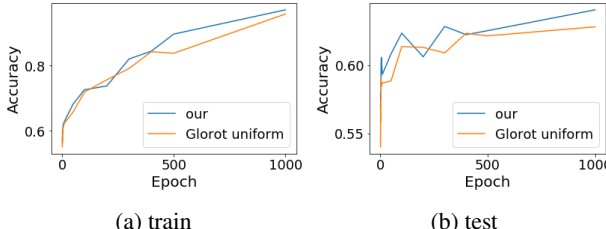

(a) train        (b) test

Figure 10: Learning curves of a fully connected linear network with one hidden layer, trained on the dogs and cats dataset, and initialized by Glorot uniform initialization (orange) and the initialization proposed in Def. 4 (blue).

Figure 11: Evaluations on test-sets projected to the first $P$ principal components, for different values of $P$ (see legend) of 10 VGG-19 models trained on CIFAR-10

### C.2.1 Spectral Bias in Classification

Rahaman et al. (2019) showed that when regressing a 2D function by a neural network, the model seems to approximate the lower frequencies of the function before its higher frequencies. Here we extend this empirical observation to the classification framework. Thus, given frequencies $\kappa = (\kappa_1, \kappa_2, ..., \kappa_m)$ with corresponding phases $\phi = (\varphi_1, \varphi_2, ..., \varphi_m)$, we consider the mapping $\lambda : [-1, 1] \to \mathbb{R}$ given by

$$\lambda(z) = \sum_{i=1}^{m} \sin(2\pi\kappa_i z + \varphi_i) := \sum_{i=1}^{m} freq_i(z) \tag{24}$$

Above $\kappa$ is strictly monotonically increasing, while $\phi$ is sampled uniformly.

The classification rule is defined by $\lambda(z) \lessgtr 0$. We created a binary dataset whose points are fully separated by $\lambda(z)$, henceforth called the *frequency dataset* (see visualization in Fig. 13 and details in §D.4). When training on this dataset, we observe that the frequency of the corresponding separator increases as learning proceeds, in agreement with the results of Rahaman et al. (2019).

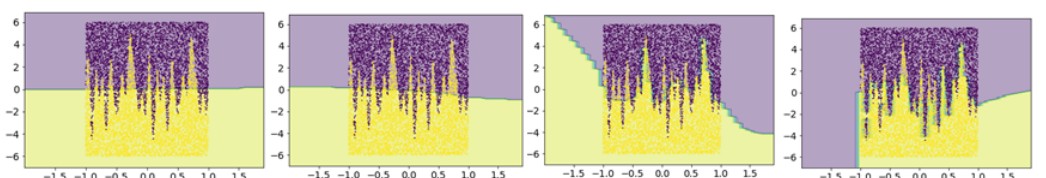

Figure 12: Visualization of the separator learned by st-VGG when trained on the frequency dataset, as captured in advancing epochs (from left to right): 1, 100, 1000, 10000. Each point represents a training example (yellow for one class and purple for the other). The background color represents the classification that the network predicts for points in that region.

To visualize the decision boundary of an st-VGG network trained on this dataset as it evolves with time, we trained $N=100$ st-VGG networks. Since the data lies in $\mathbb{R}^2$, we can visualize it and the corresponding network's inter-class boundary at each epoch as shown in Fig. 12. We can see that the decision boundary incorporates low frequencies at the beginning of the learning, adding the higher frequencies only later on. The same qualitative results are achieved with other instances of st-VGG as well. We note that while the decision functions are very similar in the region where the training data is, at points outside of the data they differ drastically across networks.

### C.2.2 Spectral Bias: Relation to Accessibility

In order to connect between the learning order, which is defined over examples, and the Fourier analysis of a separator, we define for each example its *critical frequency*, which characterizes the smallest number of frequencies needed to correctly classify the example. To illustrate, consider the *frequency dataset* defined above. Here, the *critical frequency* is defined as the smallest $j \in [m]$ such that $\lambda_j(z) = \sum_{i=1}^{j} freq_i(z)$ classifies the example correctly (see Figs. 14a,14b).

In this binary classification task, we observe a strong connection between the order of learning and the *critical frequency*. Specifically, we trained $N=100$ st-VGG networks on the *frequency dataset*,

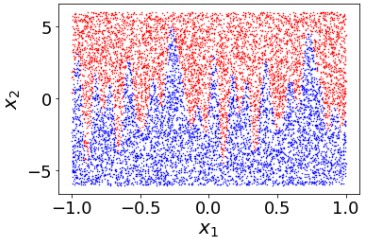

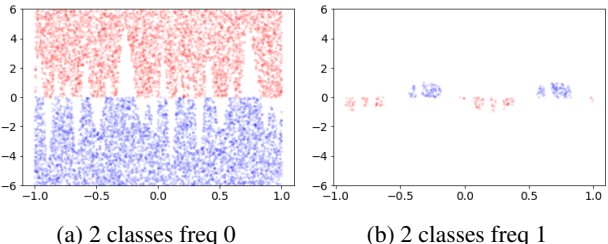

(a) 2 classes freq 0       (b) 2 classes freq 1

Figure 13: Visualization of the classification dataset used to extend Rahaman et al. (2019) to a classification framework.

Figure 14: Visualization of the *critical frequency*, showing all the points in the 2D-frequency dataset with *critical frequency* of (a) 0, and (b) 1.

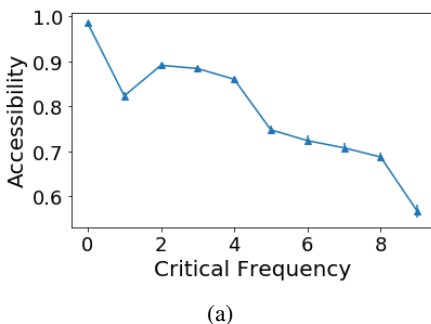

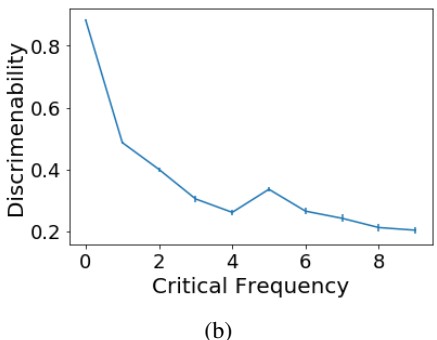

(a)               (b)

Figure 15: (a) Correlation between *critical frequency* and *accessibility* score in the 2D-frequency dataset. (b) Correlation between *discriminability* and *critical frequency* in the 2D-frequency dataset.

and correlated the *accessibility* scores with the *critical frequency* of the examples (see Fig. 15a). We see a strong negative correlation ($r = -0.93$, $p < 10^{-2}$), suggesting that examples whose *critical frequency* is high are learned last by the networks.

In order to see the effect of the *spectral bias* in real classification task and extend the above analysis to natural images, we need to define a score that captures the notion of *critical frequency*. To this end, we define the *discriminability* measure of an example - the percentage out of its $k$ neighbors that share the same class as the example. Intuitively, an example has a low *discriminability* score when it is surrounded by examples from other classes, which forces the learned boundary to incorporate high frequencies. In Fig. 15b we plot the correlation between the *discriminability* and the *critical frequency* for the 2D frequency dataset. The high correlation ($r=-0.8$, $p < 10^{-2}$) indicates that *discriminability* indeed captures the notion of *critical frequency*.

### C.3 Projection to higher PC's

In Section 3.3 we described an evaluation methodology, based on the creation of a modified *test-set* by projecting each test example on the span of the first $P$ principal components. We repeat this experiment with VGG-19 networks on CIFAR-10, and plot the results in Fig. 11.

## D Methodology

### D.1 Implementation details and hyper parameters

The results reported in Section 5 represent the mean performance of 100 st-VGG and linear st-VGG networks, trained on the small mammals dataset. The results reported in Section 5 represent the mean performance of 10 2-layers fully connected linear networks trained over the cats and dogs dataset. The results in Fig. 9 represent the mean performance of 100 st-VGG network trained on the small mammals dataset. In every experimental setup the network's hyper-parameters were coarsely grid-searched to achieve good performance over the validation set, for a fair comparison. Other hyper-parameters exhibit similar results.

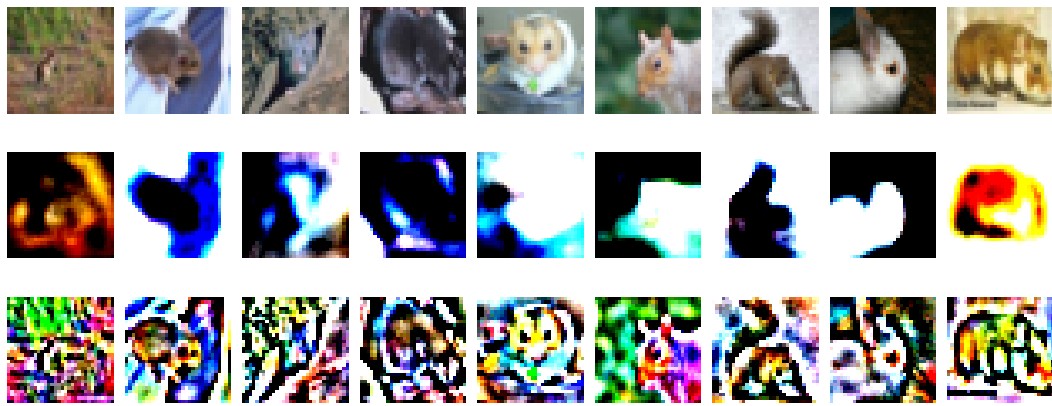

Figure 16: Visualization of the small mammals dataset, with amplification of $1.5\%$ of its principal components by a factor of 10. Top: original data; middle: data amplified along the first principal components; bottom: data amplified along the last principal components

## D.2 Generalization Gap

In Section 5 we discuss the evaluation of networks on datasets with amplified principal components. Examples of these images are shown in Fig. 16: the top row shows examples of the original images, the middle row shows what happens to each image when its $1.5\%$ most significant principal components are amplified, and the bottom row shows what happens when its $1.5\%$ least significant principal components are amplified. Amplification was done by a factor of 10, which is significantly smaller than the ratio between the values of the first and last principal components of the data. After amplification, all the images were re-normalized to have 0 mean and std 1 in every channel as customary.

## D.3 Architectures

**st-VGG.** A stripped version of VGG which we used in many of the experiments. It is a convolutional neural network, containing 8 convolutional layers with 32, 32, 64, 64, 128, 128, 256, 256 filters respectively. The first 6 layers have filters of size $3 \times 3$, and the last 2 layers have filters of size $2 \times 2$. Every other layer is followed by a $2 \times 2$ max-pooling layer and a $0.25$ dropout layer. After the convolutional layers, the units are flattened, and there is a fully-connected layer with 512 units followed by $0.5$ dropout. The batch size we used was 100. The output layer is a fully-connected layer with output units matching the number of classes in the dataset, followed by a softmax layer. We trained the network using the SGD optimizer, with cross-entropy loss. When training st-VGG, we used a learning rate of $0.05$.

**Linear st-VGG.** A linear version of the st-VGG network. In linear st-VGG, we change the activation function to the identity function, and replace max-pooling by average pooling with a similar stride.

**Linear fully connected network.** An $L$-layered fully connected network. Each layer contains 1024 weights, initialized with Glorot uniform initialization. $0.5$ dropout is used before the output layer. Networks are trained with an SGD optimizer, without momentum or $L_2$ regularization.

## D.4 Datasets

In all the experiments and all the datasets, the data was always normalized to have 0 mean and std 1, in each channel separately.

**Small Mammals.** The small-mammals dataset used in our experiments is the relevant super-class of the CIFAR-100 dataset. It contains 2500 train images divided into 5 classes equally, and 500 test images. Each image is of size $32 \times 32 \times 3$. This dataset was chosen due to its small size.

**Cats and Dogs.** The cats and dogs dataset is a subset of CIFAR-10. It uses only the 2 relevant classes, to create a binary problem. Each image is of size $32 \times 32 \times 3$. The dataset is divided to 20000 train

images (10000 per class) and 2000 test images (1000 per class). This dataset is used when a binary problem is required.

**ImageNet-20.** The ImageNet-20 dataset is a subset of ImageNet containing 20 classes. This data resembles ImageNet in terms of image resolution and data variability, but contains a smaller number of examples in order to reduce computation time. The dataset contains 26000 train images (1300 per class) and 1000 test images (50 per class). The choice of the 20 classes was arbitrary, and contained the following classes: boa constrictor, jellyfish, American lobster, little blue heron, Shih-Tzu, scotch terrier, Chesapeake Bay retriever, komondor, snow leopard, tiger, long-horned beetle, warthog, cab, holster, remote control, toilet seat, pretzel, fig, burrito and toilet tissue.

**Frequency dataset** A binary 2D dataset, used in Section 4.3, to examine the effects of spectral bias in classification. The data is define by the mapping $\lambda : [-1, 1] \rightarrow \mathbb{R}$ given in (24) by

$$\lambda(z) = \sum_{i=1}^{m} \sin(2\pi\kappa_i z + \varphi_i) := \sum_{i=1}^{m} freq_i(z)$$

with frequencies $\kappa = (\kappa_1, \kappa_2, ..., \kappa_m)$ and corresponding phases $\phi = (\varphi_1, \varphi_2, ..., \varphi_m)$. The classification rule is defined by $\lambda(z) \lesseqgtr 0$.

In our experiments, we chose $m = 10$, with frequencies $\kappa_1 = 0, \kappa_2 = 1, \kappa_3 = 2, ..., \kappa_{10} = 9$. Other choices of $m$ yielded similar qualitative results. The phases were chosen randomly between 0 and $2\pi$, and were set to be: $\varphi_1 = 0, \varphi_2 = 3.46, \varphi_3 = 5.08, \varphi_4 = 0.45, \varphi_5 = 2.10, \varphi_6 = 1.4, \varphi_7 = 5.36, \varphi_8 = 0.85, \varphi_9 = 5.9, \varphi_{10} = 5.16$. As the first frequency is $\kappa_1 = 0$, the choice of $\varphi_0$ does not matter, and is set to 0. The dataset contained 10000 training points, and 1000 test points, all uniformly distributed in the first dimension between $-1$ and $1$ and in the second dimension between $-2\pi$ and $2\pi$. The labels were set to be either 0 or 1, in order to achieve perfect separation with the classification rule $\lambda(z)$.