# OpenReview forum: "Principal Components Bias in Deep Neural Networks"
_NeurIPS.cc/2021/Conference — NeurIPS 2021 Submitted_

### Official Review · Reviewer_NiJh · 2021-07-04

**Rating:** 4
**Confidence:** 4

**Summary:**

The authors study a principal component bias of deep linear neural networks when trained with gradient descent using a small learning rate. They show that when the network is sufficiently wide, deep linear networks behave like single layer linear networks during the first phase of training, where the rate of convergence is governed by the largest principled components of the data. It is further shown that at later stages of training, under some assumptions, the PC bias remains to some extent.

**Limitations And Societal Impact:**

I would encourage the author to address the limitation detailed in the review.

**Main Review:**

My main concern with the paper is that it is overall quite incremental over previous results:
It is known for some time at this point that sufficiently wide networks evolve linearly, where convergence rate is governed by the principled components of the neural tangent kernel (NTK). For deep linear networks, the NTK itself is given by the Gram matrix of the data (\Sigma_xx using the notations in the paper) scaled by the depth L, which is identical to the NTK of a single layer linear networks up to scale. Hence, it is trivial that when the width M is large, the convergence rate in both shallow and deep linear models will exhibit the PC bias as detailed by the authors. In other words, the large width assumption when applied to linear models strips it of all the interesting properties provided by depth and relegates it to a 1 layer linear network.
For the second phase of training, i find the assumptions made by the authors quite unrealistic. I would argue that in most tasks of interest k/m cannot be considered small. Moreover, the presented results show no explicit dependency on training time, which makes it difficult to understand the regimes in which the results hold.

In general, i find the PC bias as described in the paper lacking in its ability to explain any interesting phenomena in deep learning since it is inherently present in shallow networks as well.  In addition, it does not trivially apply to deep nonlinear models. Even when only considering linear models, the incremental technical novelty in the paper does not meet the threshold for acceptance in my opinion.

Question:

1) The experimental section includes results on VGG style linear architectures, however the theoretical section does not deal with convolutions. How should the PC bias be interpreted in this case? Shouldnt the gram matrix \Sigma_xx be computed differently for convolutional architectures?


_____________________Post Rebuttal______________________

I appreciate the authors response to my review. However, my score remains unchanged for the following reasons. In my view the theoretical contribution is a restatement of known results - Theorem 2 discusses leading corrections to the infinite width limit, which have been discussed in numerous papers [1,2], and theorem 1 should be relegated to a proposition. Moreover, the actual claimed theoretical contribution - namely the O(q/m) and O(K/m) corrections, is not formally presented, and seems to be brushed aside due to technical issues (it is not clearly presented in the supplementary either).
In general i find the empirical section intriguing, however i doubt the claim of the paper carries over to larger datasets, and i feel this should be addressed in future versions of the paper.

[1] Wide Neural Networks of Any Depth Evolve asLinear Models Under Gradient Descent

[2] Asymptotics of Wide Networks from Feynman Diagrams


**Time Spent Reviewing:**

3

---

> ### Author Response · Authors · 2021-08-08
> **Response to Reviewer NiJh**
>
> Your review (both the summary and the in-depth review) ignores the empirical part of our paper, which demonstrates how the PC-bias can be found outside of the linear domain, and how it may explain several phenomena in more “realistic” networks.
>
> Our analysis proves the existence of the PC-bias in deep linear networks directly, rather than through the lens of kernel analysis. This difference is crucial for several reasons. First, the equivalence you mention only holds in the limit of infinite width - we empirically show that the PC-bias appear in practice much earlier. Second, direct proof is always preferable - while very popular, NTK still has many limitations [1], and several works question its ability to explain the successes of neural networks in difficult high dimensional tasks [2].
>
> Unrealistic assumptions:
> notations reminder: k - number of classes, m - width of the hidden layers.
>
> While k/m is not necessarily small for all problems, in practice it is usually small enough. In figure 3, we empirically show that this effect appears even in small networks (m = 1024) over CIFAR-10 (k=10), which arguably, can be considered realistic. We’ve seen similar results even with smaller networks on this data (m=512) and even on datasets with more classes, such as CIFAR-100 (k=100). We included results for CIFAR-10 only in the paper due to space limitations, but we will add them to the supplementary material in the final version.
>
>
> [1] Yehudai, Gilad, and Ohad Shamir. "On the power and limitations of random features for understanding neural networks." Advances in Neural Information Processing Systems 32 (2019): 6598-6608.‏
>
> [2] Chizat, Lenaic, Edouard Oyallon, and Francis Bach. "On Lazy Training in Differentiable Programming." NeurIPS 2019-33rd Conference on Neural Information Processing Systems. 2019.‏

---

> > ### Comment · Reviewer_NiJh · 2021-08-09
> > **Response to response**
> >
> > Thank you for the response, however i would appreciate some further clarifications.
> >
> > "First, the equivalence you mention only holds in the limit of infinite width"
> > Theorem 2 clearly states convergence as width tend to infinity. The implication of theorem 2 is that a deep infinitely wide linear network trains like a 1 hidden layer linear network. This is a direct result of the NTK theory for infinite width networks. I do not see how this is not "through the lens of kernel analysis" as you put it. Am i missing something here?
> >
> > "While k/m is not necessarily small for all problems, in practice it is usually small enough"
> > I disagree. m = 1024 is certainly wide by all standards, and cifar 10 contains merely 10 classes.  For cifar 100 on the same architecture we would have, k/m ~ 1/10. For Imagenet we would have k/m ~ 1. Even by widening further k/m could hardly be considered small.
> > Moreover i am concerned that the convergence rate paragraph in section 2.1 contains a rather vague statement ("...is governed to some extent by..."), rather than a clear theorem/lemma. What do you mean by "to some extent"?

---

> > > ### Author Response · Authors · 2021-08-09
> > > **Direct proof is more than the limit**
> > >
> > > Thanks for your speedy response!
> > >
> > > "Theorem 2 clearly states convergence as width tend to infinity. The implication of theorem 2 is that a deep infinitely wide linear network trains like a 1 hidden layer linear network."
> > > You are right. But it implies more, and specifically - that for every width m there are corresponding \epsilon and \delta such that (22) and (23) are "almost" true, and the error is infinitesimnal in a formal sense. This cannot be directly concluded from the NTK theory - it can only be stated as a hypothesis that needs to be proved (or disproved).
> > >
> > > Width. As proven in the paper, when the hidden width is fixed, convergence of the left gradient scale matrices A^l is governed by q/m (this means \frac{q}{m}), where q is the input dimension, see theorem 7 . In CIFAR10 q=3074, implying that q/m is if the order of 3 (larger than 1). In this sense m is large - the left gradient matrices are not the identity matrix, not even approximately so, and yet the qualitative behavior we see is similar (as shown in the empirical evaluation).
> > >
> > > "What do you mean by "to some extent"?" This is formally defined in Appendix A, and specifically equation (13) and Thm 7. Indeed, a clarification is need; if accepted, we will add these pointers to the discussion in section 2.1.

---

### Official Review · Reviewer_QgKK · 2021-07-08

**Rating:** 6
**Confidence:** 3

**Summary:**

The paper argues for the existence of the so-called “principal component bias” in the learning of deep wide nets, with the main focus on linear networks. This refers to the phenomenon that the learning that is associated with larger PCs of the data is typically faster. The paper does so via a theoretical analysis of the early stage of learning in linear nets and a series of experiments, involving both linear and nonlinear models.

**Limitations And Societal Impact:**

Yes

**Main Review:**

This is a paper with quite very interesting experiments. Understanding deep nets is difficult, so the ability to have a partial understanding via a simple pattern like PC bias is valuable, even though PC bias does not give the full picture as admitted in the paper. The theoretical analysis is non-rigorous and appears to be the weaker part; it is the series of small but informing experiments that give useful insights. Specifically the paper draws a nice connection between PC bias and common techniques like PC amplification, whitening and random labelling, demonstrating that PC bias can be a useful guide for subsequent studies that involve these techniques.

My main criticism lies in the theoretical part. Certain parts are not rigorous; for example:
- In the use of big O notation, this notation hides a lot of important dependencies on e.g. time, weight magnitude, etc. This has serious implication to Theorem 2; for example, if the conclusion can only hold until time $\sim O(1/m^{99})$, this implies Theorem 2 is not very useful (even though Fig. 1 is).
- The proof of Theorem 5 is actually bad: from line 515 (which is an entry-wise bound as written) to line 516, one requires a certain union bounding over $m^2$ entries, and so the correct probability bound becomes void. This would have been avoided if the metric of interest (denoted by $|\cdot |$ as in the paper) is not entry-wise absolute value, but some appropriate norm.

These can be fixed, though I think this is the less important part of the paper. Moreover previously there have been rigorous theoretical results that convey a certain sense of PC bias, by solving the complete solution of the learning dynamics, in linear networks (e.g. Gidel et al 2020) as well as nonlinear networks (e.g. Nguyen 2021), without the whitened data assumption just like this paper, but of course in tractable theoretical settings. As such, despite the weakness in the theoretical part, I would not say PC bias has no (or unreliable) theoretical basis, and I would rather emphasize on the experimental findings.

While I do not know the experimental literature sufficiently well to judge on novelty, supposing there is no issue with that and unless there are flaws in experimental designs spotted by other reviewers, I think the paper is a worthy contribution to the broader community.

References:

Gidel, Bach, Lacote-Julien, “Implicit regularization of discrete gradient dynamics in linear neural networks”, 2020.

Nguyen, “Analysis of feature learning in weight-tied autoencoders via the mean field lens”, 2021.

**Time Spent Reviewing:**

6

---

> ### Author Response · Authors · 2021-08-05
> **Response to Reviewer QgKK**
>
> We greatly appreciate your in-depth review of our paper, including the supplementary material.
>
> Theoretical part:
> Theorem 1 indeed resembles Du and Hu (2019), and we will update our paper accordingly.
>
> Our analysis is indeed different from the worst-case framework adopted in the 3 references below. The approach taken in [1-3]  focuses on the analysis of convergence bounds, while our approach is based on the notion of convergence in probability. Note that the analysis reported in [2] detects the dependence of convergence on the largest eigenvalue \lambda_max, which suitably characterizes the convergence of the upper bound,  but fails to detect more subtle dependencies of the convergence pattern which are determined by the whole spectrum of the covariance matrix. Our analysis, in contrast, reveals further properties that depend on the whole spectrum, not just \lambda_max.
>
> May we argue that work based on the bound-based approach, including [1-3], did not so far reveal the PC-bias (or similar) described in this paper. It, therefore, seems that the benefit of taking a fresh and different approach is self-evident.
>
> Empirical part:
> The theory does hold in practice. In the empirical section we change the loss and the initialization to resemble more commonly used methods, to emphasize that in practice, many of our assumptions can be relaxed. Such experiments demonstrate that our results hold in more “realistic” settings. As stated in line 182, using the L2 loss and the original initialization yield similar results. We will emphasize this point.
>
> Thank you for the helpful suggestions in improving the structure of the paper, we will address them in the final version. We will obviously reference the discussed literature where needed.
>
> [1] Du, Simon, et al. "Gradient descent finds global minima of deep neural networks." International Conference on Machine Learning. PMLR, 2019.
>
> [2] Du, Simon, and Wei Hu. "Width provably matters in optimization for deep linear neural networks." International Conference on Machine Learning. PMLR, 2019.
>
> [3] Arora, Sanjeev, et al. "Fine-grained analysis of optimization and generalization for overparameterized two-layer neural networks." International Conference on Machine Learning. PMLR, 2019.

---

> > ### Comment · Reviewer_QgKK · 2021-08-19
> > **clarification**
> >
> > Hi,
> >
> > Thanks a lot for the responses!
> >
> > It seems your responses to my review and Reviewer aYkS's have been misplaced. Can you confirm?
> >
> > With regards to the following comment:
> >
> > ```
> > As mentioned in our answer to R1, previous works such as Gidel et al 2020 assume that \Sigma_x is approximately I, and therefore their analysis cannot detect the dependency of convergence on the principal components of the data. For this reason, none reported this bias.
> > ```
> >
> > I would disagree. Both references I mentioned demonstrate (with rigorous proofs) the PC bias, though they may not state so explicitly. See for example Fig. 1 in Gidel et al 2020 and Fig. 1 in Nguyen 2021. Notice the shape of the curves.
> >
> > The difference with your work lies in the setups: Gidel et al 2020 assumes shallow linear networks, and Nguyen 2021 assumes shallow nonlinear autoencoders.

---

> > > ### Author Response · Authors · 2021-08-19
> > > **Previous work**
> > >
> > > Thank you for your attention and elaborated discussion :)
> > >
> > > You are right, our responses to you and Reviewer aYkS regarding Gidel et al 2020 have been misplaced.
> > >
> > > Gidel et al 2020:
> > > As we assume you also read the comments for Reviewer 9zUU (which asked a similar question), we will refer you there once again to see our full answer to this question (we will add a copy of our comment to Reviewer 9zUU at the end of this comment, for convenience). In brevity, we argue that on top of assuming only a different setup (2 layer linear network), Gidel et al 2020 also assume that \Sigma_x depend on the labels of the data (joint decomposition assumption), and showed bias toward the eigenvectors of \Sigma_xy instead of \Sigma_x. While under their assumptions these eigenvectors are similar, we show that in a more general way.
> > >
> > >
> > > Nguyen 2021:
> > > We did not specifically comment on this work in our initial rebuttal. However, as you mention, they indeed show that learning occurs according to the principal directions of the data, and do not assume an identity \Sigma_x. As you mention, the main difference is the shallow setup, which is different from our results. Note that they also have a very strong assumption on the data itself: they assume that the data comes from a gaussian distribution, while in our work there are no assumptions on the data itself.
> > >
> > > We agree that both papers should (and will) appear in our previous work. However, we disagree that these papers take from the novelty of our work; we view them as complementary works, finding related (but not always the same) phenomena in drastically different setups, sometimes with unrealistic assumptions.
> > > Note that our assumptions are weak enough for us to empirically observe the phenomenon even in very complex settings such as deep non-linear networks.
> > >
> > >
> > > Thank you again for your interest, we will be more than happy to provide any additional clarifications.
> > >
> > > [1] Gidel, Gauthier, Francis Bach, and Simon Lacoste-Julien. "Implicit regularization of discrete gradient dynamics in linear neural networks." arXiv preprint arXiv:1904.13262 (2019).
> > >
> > > [2] Nguyen, Phan-Minh. "Analysis of feature learning in weight-tied autoencoders via the mean field lens." arXiv preprint arXiv:2102.08373 (2021).
> > >
> > >
> > >
> > > ------------------------------------------------------------------------------------------------------------------------------------
> > >
> > >
> > >
> > >
> > > A copy of the comment for Reviewer 9zUU, for convenience:
> > >
> > >
> > > """
> > >
> > > Thank you for thoroughly discussing our work, and asking for clarifications where needed :)
> > >
> > > You are obviously correct. Unlike the previous work both our papers rely on (Saxe et al, 2018), Gidel et al (2020) do not assume it, which was an overlook on our part.
> > >
> > > However, note that both analyses are still vastly different. We have different assumptions:
> > >
> > > Gidel et al (2020) assume join decomposition of \Sigma_x and \Sigma_xy, while we do not. This is not a realistic assumption in many datasets, and it adds dependencies between \Sigma_x and the labels of the data. In our case, the learning dynamics depend on \Sigma_x alone, which is unaffected by the labels of the data at all.
> > >
> > > Gidel et al (2020) analyze a 2-layer model, while our model depth is arbitrary.
> > >
> > > Gidel et al (2020) assume vanishing initialization, while our analysis reveals how the initialization affects convergence.
> > >
> > > Most importantly, they show sequential learning of the left eigenvectors of \Sigma_xy (Section 3.1, after Thm 2). Under the joint decomposition assumption, this is indeed similar to sequentially learning the eigenvectors of \Sigma_x in agreement with our work. However, we show that this sequential learning to eigenvectors of \Sigma_x always appears, regardless of the eigenvectors of \Sigma_xy. In our work, the eigenvectors of \Sigma_xy may contribute to the speed of convergence along different directions, but the order of convergence is affected by \Sigma_x alone.
> > >
> > > """

---

> > > > ### Comment · Reviewer_QgKK · 2021-08-19
> > > > **reply**
> > > >
> > > > Thanks for the quick response!
> > > >
> > > > I agree, that as said in my review, the two references I pointed out are complementary, so as to make my point that PC bias is not without strong theoretical supports. Note that my review also points to certain weaknesses in your theoretical results / proofs, which should be edited properly. As such, my advocate for an accept is mainly owing to the interesting experimental part of the paper.
> > > >
> > > > Also do note that I may have missed other previous works, other than Gidel et al 2020 and Nguyen 2021, which may have shown the PC bias in some ways.

---

> > > > > ### Author Response · Authors · 2021-08-22
> > > > > **reply**
> > > > >
> > > > > Thank you :)
> > > > >
> > > > > Normally, we would not reply at this point, as the main points have already been mentioned. However, as our initial review score is borderline, every single point could make a difference.
> > > > >
> > > > > First, note that an argument about previous work that the reviewers and the authors are not aware of is problematic, as any such argument could always be made about any work.
> > > > >
> > > > > Secondly, as long as other papers did not explicitly found the PC bias, having strong theoretical support in other papers is a merit rather than a caveat, as it shows that our work addresses a topic that is of interest to others, and significantly contributing to it.
> > > > >
> > > > > Lastly, the element-wise bound in Theorem 5 is not wrong, but simply a shortcut: the proof does hold with the appropriate union bound (although, as you mentioned, changing this to a norm could alleviate this issue). We did not add it as the proof already is very technical, in a calculus 101 level, and adding more details felt excessive. However, we will add it for completeness. Your point on the O-notation is well taken, and you are right that if the constants are very big then the theorem is much less interesting. However, as you mentioned in your original comment, Fig. 1 shows that this is not the case in practice. We will address this issue in the final version.

---

### Official Review · Reviewer_aYkS · 2021-07-15

**Rating:** 6
**Confidence:** 4

**Summary:**

The paper studies the evolution of a deep over-parametrized linear network under gradient descent.  The main claim of the paper is that the convergence rate of the weights is faster along directions corresponding to the larger principal components of the data, at a rate governed by the singular values. The paper also supports its argument in an extensive experimental study.

**Main Review:**

The paper touches on an important question about the inductive bias of neural nets. This problem has paramount importance to the deep learning community. Moreover, the motivation of the results is clearly explained in the paper.
However, I'm not sure whether the theoretical side is strong enough to justify a new publication. Specifically:

1. Thm 1 is a direct calculation of the update at each iteration, its contribution is limited. Moreover, it seems to me like a minor modification of the update presented in "Width Provably Matters in Optimization for Deep Linear Neural Networks" by Du and Hu

2. Thm 2 (which is the main theoretical claim of the paper) presenters a new result as far as I know. However, I'm not sure about its novelty given the techniques that are used in standard over-parametrized literature (for example "Width Provably Matters in Optimization for Deep Linear Neural Networks" by Du and Hu or "Gradient descent finds global minima of deep neural networks" by Du et al). I'll be happy if the authors can address the novelty of their proof techniques and compare it to standard lazy training technique.

3. I would suggest rearranging section 2.2 to be formulated as formal theorems where the condition and results are rigorously organized. Moreover, separating more accurately between the two phases of learning might add to the paper's contribution.

4. I think it is important to include the results of the nonlinear case (section B.2 in the appendix) in the main paper. Although additional assumptions are required. These results might hint that the phenomena presented in the paper happens also outside the linear network world.

Regarding the empirical side, I think that the empirical study is impressive and definitely contributes to the paper. My main concern on the empirical side is why are the settings were changed in the empirical part? (specifically the loss and initialization). I suspect that the theory doesn't hold in practice which might weaken the paper.

To summarize my review, I didn't find major flaws in the paper. In my opinion, each part alone (the empirical and the theoretical) doesn't enough for a new publication. However, since both parts were combined together, I find the total contribution of the paper on the borderline for a top-tier conference.


**Time Spent Reviewing:**

One day

---

> ### Author Response · Authors · 2021-08-05
> **Response to Reviewer aYkS**
>
> Thank you for the elaborated review, attention to detail, and interest in our work.
>
> As none of the other reviewers found the empirical section substantially lacking, while both R1 and R3 found it novel and interesting, we hope that this major concern is considered properly addressed.
>
> While you find the theoretical part “less important”, we would argue that it further extends prior work more than you acknowledged. As mentioned in our answer to R1, previous works such as Gidel et al 2020 assume that \Sigma_x is approximately I, and therefore their analysis cannot detect the dependency of convergence on the principal components of the data. For this reason, none reported this bias.
>
> Regarding formulation: we greatly appreciate your attention to detail and will follow your advice when rewriting the relevant parts in the final version.
>
> [1] Gidel, Bach, Lacote-Julien, “Implicit regularization of discrete gradient dynamics in linear neural networks”, 2020.

---

> > ### Comment · Reviewer_aYkS · 2021-08-20
> > **Response**
> >
> > Thanks for the response.
> >
> > None of my concerns were addressed in your response.
> > Specifically, the significance of theorem 1 and theorem 2, given the work of  [Du and Hu]  (which hasn't been cited in your work) was not explained. As I mention in my comments- the main theorems of this work might be a straightforward extension of  [Du and Hu]. This is a major comment that shouldn't be overlooked.
> >
> > Another concern of mine which was not addressed is the setting of the empirical section- " My main concern on the empirical side is why are the settings were changed in the empirical part? (specifically the loss and initialization). I suspect that the theory doesn't hold in practice which might weaken the paper." Please also address this concern.
> >
> >
> > "Width Provably Matters in Optimization for Deep Linear Neural Networks" by Du and Hu

---

> > > ### Author Response · Authors · 2021-08-20
> > > **Mixup**
> > >
> > > Unfortunately, we misplaced our comments - the comment you see above was meant to answer the concerns of Reviewer QgKK, while the comment we originally wrote for you was written as a response to Reviewer QgKK.
> > >
> > > Thank you for not giving up on us when we supplied the wrong comment. This is not trivial, and we greatly appreciate it.
> > >
> > > We will recite the original comment we wrote you (also could be found under QgKK section):
> > >
> > > -------------------------------------------------------------------------
> > >
> > > We greatly appreciate your in-depth review of our paper, including the supplementary material.
> > >
> > > Theoretical part: Theorem 1 indeed resembles Du and Hu (2019), and we will update our paper accordingly.
> > >
> > > Our analysis is indeed different from the worst-case framework adopted in the 3 references below. The approach taken in [1-3] focuses on the analysis of convergence bounds, while our approach is based on the notion of convergence in probability. Note that the analysis reported in [2] detects the dependence of convergence on the largest eigenvalue \lambda_max, which suitably characterizes the convergence of the upper bound, but fails to detect more subtle dependencies of the convergence pattern which are determined by the whole spectrum of the covariance matrix. Our analysis, in contrast, reveals further properties that depend on the whole spectrum, not just \lambda_max.
> > >
> > > May we argue that work based on the bound-based approach, including [1-3], did not so far reveal the PC-bias (or similar) described in this paper. It, therefore, seems that the benefit of taking a fresh and different approach is self-evident.
> > >
> > > Empirical part: The theory does hold in practice. In the empirical section we change the loss and the initialization to resemble more commonly used methods, to emphasize that in practice, many of our assumptions can be relaxed. Such experiments demonstrate that our results hold in more “realistic” settings. As stated in line 182, using the L2 loss and the original initialization yield similar results. We will emphasize this point.
> > >
> > > Thank you for the helpful suggestions in improving the structure of the paper, we will address them in the final version. We will obviously reference the discussed literature where needed.
> > >
> > > [1] Du, Simon, et al. "Gradient descent finds global minima of deep neural networks." International Conference on Machine Learning. PMLR, 2019.
> > >
> > > [2] Du, Simon, and Wei Hu. "Width provably matters in optimization for deep linear neural networks." International Conference on Machine Learning. PMLR, 2019.
> > >
> > > [3] Arora, Sanjeev, et al. "Fine-grained analysis of optimization and generalization for overparameterized two-layer neural networks." International Conference on Machine Learning. PMLR, 2019.
> > >
> > > ------------------------------------------------------------
> > >
> > > We would like to add that while the analysis in [Du and Hu] had a different focus and notation than ours, proving Theorem 1 in our paper is indeed much simpler given Section 5 of their paper. We already revised our work to include it as a proposition followed by their work. However, Theorem 2, which analyses the gradient scale matrices in-depth, and the entire weight evolution section (and the results on the non-linear case in appendix b.2) are entirely new and required substantial further analysis above what was done in [Du and Hu].
> > >
> > > Empirical section: The entire theory holds in practice. The results hold both for the original settings and the more "realistic" ones, and we chose to present only the "realistic" as in the given space, we could not show them both. As the original settings were already proven to work, it seems of greater significance to show the new settings as well. This point is written explicitly in the current version of the paper, but we will make it clearer in future versions.

---

> > > > ### Comment · Reviewer_aYkS · 2021-09-07
> > > > **Response**
> > > >
> > > > Thanks for the response. My view didn't change much, I still think that this work is a borderline. Some of the theoretical results are not new and there are several weaknesses in both the theoretical and empirical sides. After reading the other reviews and the author's response my rating stays borderline and now I'm inclined to reject.

---

> > > > > ### Author Response · Authors · 2021-09-07
> > > > > **novelty of results**
> > > > >
> > > > > Can you please be more specific about which of the theoretical results "are not new"?  The main result of section 2.1 is summarized in theorem 2; the main results of section 2.2 are summarized in (4) and (5); the mathematics needed to achieve these results is provided in Thms 3-7 in Appendix A. In order to dispute the novelty of these results, references to earlier reports are required.

---

> > > > > > ### Comment · Reviewer_aYkS · 2021-09-07
> > > > > > **Response**
> > > > > >
> > > > > > "Some results are not new" means for theorem 1 which can be cited from Du and Hu. As I mentioned in my review, as far as I know, theorem 2 is new and I don't doubt it.

---

> > > > > > > ### Author Response · Authors · 2021-09-08
> > > > > > > **Agreed summary: 1 theorem, out of 7 theorems and 3 lemmas, is not new**
> > > > > > >
> > > > > > > Thereom 1 is there in order to arrive at the main result of section 2.1, namely, Thm 2. The main theoretical results of the paper, which are being evaluted empirically, are actually shown in section 2.2. The main theoretical novelty is detailed in appendix A. To say that "some of the theoretical results are not new" regarding a paper with 7 theorems and 3lemmas, where the only overlap lies in the first theorem, which could have been called a lemma (as it is there only to be used in proving the main result), is hardly a fair summary.

---

### Official Review · Reviewer_9zUU · 2021-07-17

**Rating:** 5
**Confidence:** 3

**Summary:**

This work studies the training dynamics of over-parameterized deep linear networks. The authors propose the Principal Components bias (PC-bias) convergence pattern that characterizes the convergence behavior of deep linear networks training, which is supported by theoretical analysis and empirical results. This works also investigates the Learning Order Constancy effect (LOC-effect) and identified the connection between PC-bias and LOC-effect. Empirically, the authors also study several implications of PC-bias, including early stopping, convergence behavior under label noise, and provide interesting observations.

**Ethical Concerns:**

N.A.

**Limitations And Societal Impact:**

N.A.

**Main Review:**

This paper identifies several interesting phenomenons for deep (non-)linear neural networks, including the effect of whitened data, connections between the PC-bias, LOC-effect, and spectral bias.
Pros:
1. Theoretically, this paper provides precise characterizations of the training dynamics of deep linear networks in Section 2.
2. Empirically, the authors investigated the proposed PC-bias in several settings as well as its connections to LOC-effect and spectral bias.

Cons:
1. The theoretical and empirical results are somehow limited to simple linear models, and some of the interesting observations have been studied in previous works, for example, [1] studied the dynamics of deep linear network training and found that the 'networks sequentially learn the solutions of a reduced-rank regression with a gradually increasing rank'.
2. The connection between whitened data, label noise, and convergence speed is interesting, it would be better to precisely characterize this phenomenon in a lemma/theorem by making some reasonable assumptions.

[1]. Implicit Regularization of Discrete Gradient Dynamics in Linear Neural Networks. Gauthier Gidel, Francis Bach, and Simon Lacoste-Julien, NeurIPS 2019.

**Time Spent Reviewing:**

4

---

> ### Author Response · Authors · 2021-08-05
> **Response to Reviewer 9zUU**
>
> Thank you for your review.
>
> Cons:
> 1 . Theoretical contribution: our analysis goes beyond the analysis of Gidel et al. (2020), Arora et al. (2019), and Saxe et al. (2019), where the PC-bias is essentially eliminated by the prior assumptions made in all 3 papers (and others). Thus such analysis cannot detect the PC bias.
>
> Specifically, we do not assume that the raw data covariance matrix \Sigma_x is approximately I, nor do we assume vanishing initialization. Our results show that models converge according to the principal components of \Sigma_x, a phenomenon that cannot be detected when assuming that \Sigma_x is the identity matrix. Please recall that all 3 aforementioned references assume exactly this - that \Sigma_x is the identity matrix. Finally, we note that eliminating the assumption of vanishing initialization further generalizes our findings.
>
> While this point is briefly mentioned in the paper (4’th paragraph, staring at line 32), we will better emphasize it, adding additional references to the previous work section.
>
> Note that the empirical results are not limited to the linear case, and even the theoretical results can be extended to simple ReLU models (see supplementary B.2), although with unrealistic assumptions.
>
> 2. In the linear case, all 3 observations about whitened data, label noise, and convergence speed can be deduced as corollaries from our analysis. We will gratefully take your advice and state them more explicitly in the paper.
>
> [1] Gidel, Gauthier, Francis Bach, and Simon Lacoste-Julien. "Implicit regularization of discrete gradient dynamics in linear neural networks." arXiv preprint arXiv:1904.13262 (2019).
>
> [2] Arora, Sanjeev, et al. "Fine-grained analysis of optimization and generalization for overparameterized two-layer neural networks." International Conference on Machine Learning. PMLR, 2019.
>
> [3] Saxe, Andrew M., James L. McClelland, and Surya Ganguli. "A mathematical theory of semantic development in deep neural networks." Proceedings of the National Academy of Sciences 116.23 (2019): 11537-11546.

---

> > ### Comment · Reviewer_9zUU · 2021-08-19
> > **Clarification question**
> >
> > Thanks for the response.
> >
> > **About the raw data covariance matrix $\Sigma$ assumption.** As mentioned in Assumption 1 (in [1]), [Gidel et al., 2020] assume the covariance matrix $\Sigma_x$ has the joint decomposition with $\Sigma_{xy}$ , and do not assume $\Sigma_{x}$ is approximately an identity matrix. Please correct me if I'm wrong.
> >
> >
> > [Gidel et al., 2020] Gidel, Gauthier, Francis Bach, and Simon Lacoste-Julien. "Implicit regularization of discrete gradient dynamics in linear neural networks." arXiv preprint arXiv:1904.13262 (2019).

---

> > > ### Author Response · Authors · 2021-08-19
> > > **Previous work**
> > >
> > > Thank you for thoroughly discussing our work, and asking for clarifications where needed :)
> > >
> > > You are obviously correct. Unlike the previous work both our papers rely on (Saxe et al, 2018), Gidel et al (2020) do not assume it, which was an overlook on our part.
> > >
> > > However, note that both analyses are still vastly different. We have different assumptions:
> > > - Gidel et al (2020) assume join decomposition of \Sigma_x and \Sigma_xy, while we do not. This is not a realistic assumption in many datasets, and it adds dependencies between \Sigma_x and the labels of the data. In our case, the learning dynamics depend on \Sigma_x alone, which is unaffected by the labels of the data at all.
> > >
> > > - Gidel et al (2020) analyze a 2-layer model, while our model depth is arbitrary.
> > >
> > > - Gidel et al (2020) assume vanishing initialization, while our analysis reveals how the initialization affects convergence.
> > >
> > > Most importantly, they show sequential learning of the left eigenvectors of \Sigma_xy (Section 3.1, after Thm 2). Under the joint decomposition assumption, this is indeed similar to sequentially learning the eigenvectors of \Sigma_x in agreement with our work. However, we show that this sequential learning to eigenvectors of \Sigma_x always appears, regardless of the eigenvectors of \Sigma_xy. In our work, the eigenvectors of \Sigma_xy may contribute to the speed of convergence along different directions, but the order of convergence is affected by \Sigma_x alone.
> > >
> > > We will extend our previous work section to include this discussion. Thanks!
> > >
> > > We will be more than happy to provide any additional clarifications.
> > >
> > >
> > > [1] Gidel, Gauthier, Francis Bach, and Simon Lacoste-Julien. "Implicit regularization of discrete gradient dynamics in linear neural networks." arXiv preprint arXiv:1904.13262 (2019).
> > >
> > > [2] Saxe, Andrew M., James L. McClelland, and Surya Ganguli. "A mathematical theory of semantic development in deep neural networks." Proceedings of the National Academy of Sciences 116.23 (2019): 11537-11546.

---

> > > > ### Comment · Reviewer_9zUU · 2021-08-19
> > > > **Quick response**
> > > >
> > > > Thanks for the quick response! This clarifies my previous question.

---

### Decision · Program_Chairs · 2021-09-27

**Decision:**

Reject

**Comment:**

Reviews of this paper were mixed. On the upside, several reviewers commended the insightfulness of its empirical investigation.  On the other end, concerns were raised regarding the novelty and significance of its theoretical contribution in light of known results in the ultra width (NTK) regime.  The latter point does not seem to be clearly delineated in the text.  Overall, while this work clearly has potential to be of interest to the community, due to the aforementioned drawback, it currently falls short of the (extremely competitive) NeurIPS acceptance threshold. I encourage the authors to clearly distill the novelty of their theory, reconsider how much weight is allocated to this theory in light of the distilled novelty, modify the text accordingly, and resubmit to a future conference.